# Determinants of Green Purchase Intention: The Roles of Green Enjoyment, Green Intrinsic Motivation, and Green Brand Love

**Yu-Hsien Lin**

Department of Urban Industrial Management and Marketing, University of Taipei, Taipei 10048, Taiwan; drlinyh@go.utaipei.edu.tw

**Abstract:** This study investigated the relationship among green enjoyment, green brand love, green intrinsic motivation, and green purchase intention. Data were collected from 26 August to 16 September 2022, through a questionnaire survey distributed online, and quantitative instruments were applied to analyze the data. A total of 302 randomly selected samples from consumers with experience of green consumption were analyzed. The data were analyzed using descriptive statistics and confirmatory factor analysis. The results revealed that the content, discriminant, and convergent validity and reliability of the model were satisfactory. Global model analysis of green intrinsic motivation revealed acceptable results. Moreover, structural equation modeling indicated a satisfactory model fit to the standard sample data. Finally, the study revealed that green intrinsic motivation positively influences green enjoyment, green brand love, and green purchase intention. Green enjoyment positively affects green brand love and green purchase intention. Furthermore, green enjoyment and green brand love mediate the positive relationship between green intrinsic motivation and green purchase intention.

**Keywords:** green intrinsic motivation; green enjoyment; green brand love; green purchase intention

## 1. Introduction

Climate change is a global problem with drastic effects, including changing weather patterns, extreme weather, food shortages, and natural resource depletion. Therefore, implementing measures for curbing climate change is imperative. The Paris Agreement was signed in 2015 as part of international efforts to curb climate disasters and global warming. Moreover, global action is being taken in response to climate-related disasters and climate change. For example, the fifth session of the United Nations Environment Assembly—Which was held in Nairobi, Kenya, from 28 February to 2 March 2022—Gathered representatives of the 193 United Nations member states to begin the process of formulating a landmark treaty aimed at reducing plastic pollution worldwide. Environmentally friendly behaviors are also crucial for alleviating the negative impacts of human activities on the environment and have thus been promoted. The purchase of environmentally friendly products and services is an emerging trend. Green consumption is integral to sustainable business, nature preservation, and environmental hazard prevention in environmental protection; it must also achieve the goals of satisfying existing consumers and attracting new consumers. Many consumers derive personal satisfaction from living frugally and engaging in environmental protection activities. Consequently, such consumers are willing to purchase green products and services on the basis of their personal values.

For 2020–2025, the UK Conservative–Liberal Democrat government implemented two key policies. First, the UK government would support innovation that increases the environmental friendliness of products and services. Second, the UK government would encourage resource efficiency and environmental management [1]. The environment–economy nexus is a prominent topic among large companies. The manufacturing industry has placed increasing emphasis on producing environmentally friendly products because

of increasing pressure from consumers, environmental activists, and government regulators. The ubiquity of online customer shopping has exerted similar pressure on the e-commerce industry to embrace environmentally conscientious supply chain practices. To meet customer demands for environmentally friendly and sustainable products, firms must establish green-oriented management strategies that emphasize sustainability [2]. With environmental regulations and increasing liability for producing hazardous products, firms are increasingly motivated to incorporate green practices in their operations. Manufacturers are eager to implement strategies such as reverse logistics and recycling to reduce environmental hazards and costs. Firms may be able to effectively combine existing and newly acquired environmental knowledge to gain a competitive advantage over their competitors in e-commerce. Environmental topics such as energy efficiency are also gaining momentum. Numerous firms have complied with environmental regulatory standards by adopting the market-leading Carbon Trust Standard and attaching carbon footprint labels to their products. Environmental and consumer advocates believe in preserving the planet and protecting the environment, and firms must consider these groups when making decisions on green products to improve their performance.

The environment is changing, and in response, firms must use up-to-date information to develop the techniques and secure the resources required to maintain and expand their business operations. Environmental responsibility has become a major topic in business. Consumers increasingly favor and are willing to pay a premium for eco-friendly products [2]. Environmental ethics and a sense of environmental responsibility influence how consumers view businesses and their products and services. Consumers are becoming more likely to make ethical or sustainable choices in purchasing situations. Modern consumers think about, believe in the power of, and want to make ethical decisions; therefore, they actively reduce their harmful consumption behaviors. Accordingly, compared with other factors, firms' green marketing strategies for their green products are crucial for gaining market advantage [3–5].

Intrinsic motivation is autonomously activated when people engage in behaviors or activities from which they derive inherent satisfaction, rather than deriving satisfaction by achieving a particular outcome. However, how intrinsic motivation can be applied to eco-friendly living and whether any connection exists between intrinsic motivation and enjoyment in the green context require clarification. If people could satisfy their hedonic needs by demonstrating eco-friendly behaviors, they would be driven by antecedents relating to environmentally friendly behaviors. Marketing managers can thus boost consumers' desire for and dedication to green living and green consumption by offering environmentally friendly products or services. The positive feedback that people receive from engaging in green consumption enables them to meet their own ethical standards relating to environmental protection.

Not all firms are capable of adding value to their products and services, particularly products and services that are claimed to be environmentally friendly. Consequently, consumers are prone to disbelieving the green marketing claims of firms [6]. Firms must implement mechanisms for producing green products and services and establish guidelines for attracting consumers to purchase such products and services. If firms could use environmental factors to their advantage, their products and services would be warmly received by consumers. A growing number of firms have certified themselves and their products and services as energy efficient and carbon neutral. However, this measure is insufficient and ineffective if firms do not add pleasing or hedonic elements to their products and services. Thus, industries, instead of aiming to enhance consumer enjoyment, must implement environmentally friendly measures to promote "green enjoyment" (GE). As reported by Davis, Bagozzi, and Warshaw [7], enjoyment is a pleasant feeling, and firms must implement strategies to enable consumers to derive enjoyment from their products or services in an environmentally friendly manner. Firms must explore different environmentally friendly factors to benefit from the green consumption trend.

Green consumption is a major aspect of sustainable behavior and is an effective approach to mitigating the negative effect of human activities on the environment. Purchasing environmentally friendly products and services can generate feelings of satisfaction among environmentally conscious consumers [8,9], enhancing the general consumption experience. When consumers perceive themselves as having low social worth, the positive effect of purchasing green products as part of the consumption experience is amplified [10]. Accordingly, this study introduced the new concept "green enjoyment" (GE), which is a type of intrinsic benefit people can derive from engaging in environmentally friendly behaviors. However, questions that remain are whether intrinsic motivation is related to GE and whether the research framework is still reasonable when intrinsic motivation and GE are incorporated. This study proposed that GE is crucial for green consumption and can both increase corporate revenue and support sustainability goals.

Green brand love (GBL) is an increasingly critical topic in business. As part of the trend of forming long-term relationships with consumers, brand love is prevalent in many different industries. Large companies such as American Eagle Outfitters, Aeropostale, Express, J.Crew, and H&M are contemplating launching brand love initiatives [11]. Moreover, more companies are selling products and services labeled as environmentally friendly, leading to another growing trend. In the 2009 Grocery Manufacturers Association (GMA)/Deloitte Consulting study, GMA noted a major opportunity for companies to meet latent demands for green products. Socially responsible or green goods and services are becoming increasingly important for retailers, and their market presence continues to grow rapidly [12]. Accompanied by appropriate regulation, trade can assist the transition to a green economy by encouraging the exchange of environmentally friendly goods and services. Tully and Winer [12] observed that over half of all the participants in their study (60.1%) were willing to pay a premium for such products. Laroche, Michel, Bergeron, Jasmin, and Barbaro-Forleo Guido [13] reported that 80% of consumers who were inclined to pay a premium for green products refused to purchase from companies that do not follow environmental regulations or that misrepresent nongreen products as green. Firms typically announce goals to protect the environment and reduce environmental hazards to both satisfy existing consumers and attract new ones. In terms of personal values, people who actively participate in various environmental protection activities derive satisfaction from living frugally and engaging in environmental protection. Consequently, such consumers are willing to purchase green products or services, given their environmental concerns and personal values [14]. Many consumers thus tend to purchase green products in the environmental era [15,16]. Some companies such as Tesla, Apple, IKEA, and Johnson & Johnson already manufacture products made of recycled materials to actively promote GBL. Because of its increasingly prominent role, studies have applied GBL as an antecedent construct [17] and outcome construct [17,18], but they have failed to explore the effect of GBL on green purchase intention (GPI). The practical contributions of the current study are as follows: green brand love is rarely analyzed with respect to its antecedent and outcome variables. Understanding green brand love in purchase behaviors could be helpful for brand managers in developing stronger brands. The proposed green purchase intention model also proves that green brand love increases the effect of brand engagement on purchase attitude. The theoretical contributions of the current study are as follows: the idea of increasing green intrinsic motivation, green enjoyment, green brand love, and green purchase intention represents a new strategy. Moreover, the present study proposed the new constructs green enjoyment and analyzed the relationships among green intrinsic motivation, green enjoyment, green brand love, and green purchase intention to fill the literature gap.

## 2. Literature Review and Hypothesis Development

The 2015 United Nations Climate Change Conference was held in Paris in December 2015. According to the Paris Agreement, the signing parties would "pursue efforts to" limit the global temperature increase to 1.5 °C. According to the United Nations' Intergovernmental Panel on Climate Change, meeting the 1.5 °C goal requires reducing carbon

emissions to zero by 2030 to 2050. In the context of global environmental protection initiatives and consumers' growing environmental awareness, firms can promote GE and green brand love (GBL) through their products and services to generate both environmental and financial benefits. With consumers becoming more sensitive to the environmental impact of their purchasing behaviors, brands must extend beyond claims of eco-friendliness and identify new environmental elements to attract consumers. Consumers want to associate products with enjoyment and as a reflection of them and their behaviors. Therefore, GE and GBL result from consumers' identification with a product or service. Accordingly, the present study explored the crucial elements in consumers' emotional connection to brands.

## 2.1. Positive Effect of GIM on GE

Studies have often measured intrinsic motivation in a unidimensional manner and have regarded high intrinsic motivation as indicative of low extrinsic motivation [19–21]. Deci and Ryan [22] defined intrinsic motivation as the execution of an activity because of its inherent appeal and satisfaction. Therefore, intrinsic motivation can be described as a desire to perform an activity or behavior or to achieve a standard. In contrast to intrinsic motivation, extrinsic motivation refers to the execution of an activity under the influence of an external force. Harter [23] studied the learning motivation of high school students in four states in the United States; he divided motivation into extrinsic and intrinsic motivation and proposed that students would engage in an academic task for intrinsic or extrinsic reasons, or both. He used 16 items to assess each motivation type on separate subscales. The results revealed that intrinsic motivation decreased from Grade 3 to Grade 9. Intrinsic motivation encourages people to perform an action because of its own inherent characteristics, whereas extrinsic motivation encourages people to perform an action to achieve a separable outcome, such as receiving money or fame [24]. Intrinsically motivated behaviors are those performed without any reward or a valued consequence. Deci et al. [24] reported that people attribute less intrinsic motivation to individuals who receive greater rewards for an activity than they do to those who receive fewer rewards. Furthermore, Fuchs, Schreier, and van Osselaer [25] reported that handmade products are given more favorable evaluations, because the producers were perceived as having intrinsic (rather than extrinsic) motivation. Li et al. [26] stated that green intrinsic motivation (GIM) is operative when people or employees perceive their locus of causality to be internal. The current study referred to the aforementioned explanation of intrinsic motivation and applied a six-item scale to measure GIM. GIM reflects an individual's tendency to engage in proenvironmental behaviors as well as their interest in, curiosity toward, and self-expression related to green products. For example, a person who purchases more environmentally friendly products because they perceive the products as interesting or because they believe that they can derive satisfaction from performing this proenvironmental behavior rather than perceiving the product as being of value to them is demonstrating the behavior on the basis of intrinsic motivation rather than for extrinsic reasons.

Enjoyment is conceptually linked to motivation through positive feelings associated with performing an activity when intrinsically motivated [27]. Some scholars have asserted that enjoyment is an attitude [28] or experiential state [29], whereas others have argued that the definition of enjoyment is rooted in intrinsic motivation [29,30]. Waterman [31] stated that "hedonic enjoyment may be expected to be felt whenever a pleasant effect accompanies the satisfaction of needs, whether physical, intellectual, or socially based" (p. 679). Enjoyment is the satisfaction of both hedonic and nonhedonic intrinsic needs [32], whereby people perceive the behavior itself to be intrinsically interesting or pleasurable. Studies have yet to explore the relationship between GIM and GE. Accordingly, the present study was conducted to fill this research gap. On the basis of the definition of enjoyment provided by Tamborini et al. [32], the present study defined GE as the satisfaction of both hedonic and nonhedonic needs through the execution of eco-friendly behaviors. This study applied a three-item scale to measure GE. When consumers engage in an environmentally friendly behavior, they experience self-endorsement of their actions, behaviors, and volition.

Therefore, when consumers are intrinsically motivated by green products or services, they are likely to enjoy the experience of expressing their positive feelings about sustainability and eco-friendliness. Through GIM, the purchase behavior itself is rewarding, the consumption situation is pleasurable, and GE is experienced. Therefore, this study proposed the following hypothesis:

**Hypothesis 1 (H1).** *GIM positively influences GE.*

### 2.2. Positive Effect of GIM on GBL

Intrinsic motivation is characterized by personal investment and engagement [33]. Li et al. [26] asserted that with intrinsic motivation, this investment is based on an interest in and passion for preserving and caring for the environment. Environmental ethics underlie GIM, which in turn drives the purchasing of environmentally friendly products and services. Li et al. [26] noted that GIM reflects an employee's inherent interest in, love for, passion about, enjoyment of, or satisfaction from proenvironmental behavior. Ali, Ashfaq, Begum, and Ali [34] conducted a study in three cities in China to investigate young Chinese consumers' behaviors toward purchasing electronic products. They determined that GIM has a positive effect on GPI. However, evidence indicating whether GIM affects GBL is lacking. The present study thus explored the relationship between GIM and GBL to fill the research gap.

Brand love refers to consumers' love for and attachment to a specific brand [35–37] and was first empirically studied by Ahuvia [38] in 1993. Fournier [39] stated that love is a core element of consumers' relationships with brands. Brand love is a fundamental element for forming brand loyalty and can be regarded as an antecedent of brand loyalty. Brand love can be regarded as a relationship between consumers and brands rather than an emotion itself [40]. Additionally, Park, MacInnis, Priester, Eisingerich, and Iacobucci [41] used the term "brand love emotion" to refer to the specific affective state generated through the consumer–brand relationship. Although the definition of brand love is clear and has been mentioned by numerous scholars, the numbers of dimensions and the measurement methods for brand love vary. Researchers have identified several dimensions of brand love, ranging from 1 (e.g., [42]) to 11 dimensions [43]. Carroll and Ahuvia [42] and Bergkvist and Bech-Larsen [44] have both maintained that brand love is a unidimensional construct. Batra et al. [40] stated that brand love consists of three factors: passion-driven behaviors, self–brand integration, and positive emotional connection. Moreover, Verma [45] proposed that the concept consist of the following factors: high quality, emotion, and passion. Batra et al. [40] argued that brand love, as consumers experience it, is best represented as a higher-order construct that includes multiple cognitions, emotions, and behaviors, which consumers organize into a prototype or cognitive reference point. The present study defined GBL as the intensity of positive emotions and enthusiasm consumers have toward a green brand. Salehzadeh, Sayedan, Mirmehdi, and Aqagoli [18] asserted that GBL is inherently dynamic and changes over time. Salehzadeh et al. [18] developed a seven-item scale for measuring GBL; the present study used this scale as a reference to design a seven-item GBL scale. Love for the environment is an essential element of GIM. Consumers who feel more intrinsically motivated to purchase environmentally friendly products likely have a higher degree of GBL. Accordingly, this study proposed the following hypothesis:

**Hypothesis 2 (H2).** *GIM positively influences GBL.*

### 2.3. Positive Effect of GIM on GPI

Intrinsic motivation refers to an inner force that drives an individual's behavior. Shang, Chen, and Shen [46] used extrinsic motivation and intrinsic motivation to assess consumer online shopping behavior. They determined that extrinsic motivation consists of only one factor, namely, perceived usefulness, whereas intrinsic motivation consists of two factors,

namely, perceived ease of use and cognitive absorption; their results revealed that intrinsic motivation has positive effects on online shopping. Kim, Lee, and Bonn [47] studied travel-related purchase intention among older adult users of mobile social networking sites. They determined that extrinsic motivation consists of the factors of usefulness and social interaction and that intrinsic motivation consists of enjoyment and self-efficacy; their results indicated that intrinsic motivation has a positive effect on purchase intention.

Dodds, Monroe, and Grewal [48] studied the effects of price, brand, and store information on buyers' perceptions of product quality and value as well as their willingness to buy. They used a five-item scale to assess purchase intention. They reported that price, brand, and store information affect perceived quality, perceived sacrifice, and perceived value as well as consumers' willingness to buy. Chen and Deng [49] defined green purchase intention (GPI) as an individual's readiness to demonstration green purchase behaviors, mainly in consideration of pollution reduction. Li et al. [26] also reported that individuals with high GIM tend to exhibit eco-friendly behaviors. On the basis of Chen and Deng's [49] definition of GPI, the present study developed a four-item scale to measure GPI.

The present study posited that consumers' values and interests determine whether they are likely to distinguish between environmentally friendly and environmentally unfriendly products, and hence, whether their GIM engenders an intention to reduce pollution through their purchase behavior. Environmentally unfriendly products include those that do not use recyclable components or appropriate materials for packaging and those manufactured using legal or illegal harmful substances. In contrast, environmentally friendly products include those that use natural substances and sustainable materials and that reduce adverse environmental effects. Lastovica, Bettencourt, Hughner, and Kuntze [50] noted that frugal and eco-centric motivations have positive effects on consumers' product use behavior. People with high GIM would be expected to purchase eco-friendly products and services. Therefore, this study proposed the following hypothesis:

**Hypothesis 3 (H3).** *GIM positively influences GPI.*

*2.4. Positive Effect of GE on GBL*

Waterman [51] stated that feelings of personal expressiveness arise when a person is in the process of self-realization through the fulfillment of their personal potentials, when personal potentials take the form of the "development of one's skills and talents, the advancement of one's purpose in living, or both" (p. 679). Ryan, Rigby, and Przybylski [52] conducted a study of video game players and reported that they experienced enjoyment through expressions of competency and autonomy while playing. On the basis of the theory presented by Ryan et al. [52], Tamborini et al. [32] determined that enjoyment is an experience that includes the satisfaction of hedonic and nonhedonic needs. Tamborini et al. [32] referred to Vorderer's [53] two-factor model of media enjoyment when testing their model of hedonic and nonhedonic needs in video game players and interactive video game players. Vorderer's model includes a lower-order factor labeled enjoyment and a higher-order factor labeled appreciation. Vorderer [53] indicated that enjoyment and appreciation could be satisfied separately through exposure to diverse forms of media entertainment. Tamborini et al. [32] developed an enjoyment model by replacing the terms "lower-order" and "higher-order" with the terms "hedonic" and "nonhedonic" to examine how the hedonic (arousal and absorption) and nonhedonic (competence, autonomy, and relatedness) components of need satisfaction are related to each other in the context of media entertainment. They used three- and seven-item scales to measure enjoyment in video game players and interactive video game players, respectively; their findings provide valuable insight into need satisfaction and extend the scholarly understanding of enjoyment and its conceptualization. According to the aforementioned literature, GE can be regarded as the satisfaction of hedonic and nonhedonic needs rooted in eco-friendly behaviors. Enjoyment is still a major topic in the field of marketing, particularly in relation to green consumption. For example, Ford offers mass customization options for online consumers,

thereby increasing consumers' enjoyment when they are choosing products. To reduce costs and increase customer responsiveness, firms may develop products that are perceived as environmentally friendly and that can be used to increase GE. Green consumers tend to develop strong relationships with trustworthy brands that provide green products or services. Furthermore, if others perceive green consumers as enjoying green consumption and as gaining satisfaction through interaction with the environment, they may also be attached to an eco-friendly brand. Engaging in an environmentally friendly behavior can also elicit positive emotions through the generation of a positive self-image. Through their relationship with brands, consumers gain opportunities to construct and maintain their identity and to achieve feelings of love and attachment. The more GE they experience, the more positive emotions and enthusiasm they feel. Understanding how to initiate, develop, and maintain high quality consumer–firm relationships is critical for business success [54].

Carroll and Ahuvia [42] argued that brand love has a much stronger affective focus than satisfaction does. Brand love is influenced by various factors, including intrinsic motivation, passion-driven behaviors, and self–brand integration. GBL reflects consumers' positive emotion toward and attachment to a brand that emphasizes sustainability and eco-friendly business practices through its products and services. People who have positive environmentally friendly purchasing experiences have a greater willingness to declare their love for green brands. Their GE experiences increase their preference for and confidence in the products and services of brands that do not harm the environment. Accordingly, this study proposed the following hypothesis:

**Hypothesis 4 (H4).** *GE positively influences GBL.*

### 2.5. Positive Effect of GE on GPI

Xu, Chen, Peng, and Anser [55] investigated the behavior of consumers engaging in online shopping in China. They reported that certain game elements affect consumers' enjoyment and psychological need satisfaction, thus influencing their online purchase intention. Furthermore, Wang, Yeh, and Liao [56] analyzed consumer intention to purchase online content services; they determined that perceived enjoyment directly influences purchase intention. Enjoyment is experienced when needs, including physical, intellectual, and social, are satisfied [51]. GE is a form of happiness that generates hedonic satisfaction through purchase of green products or services. Purchasing is not only about obtaining products or services but is also an enjoyable experience and a form of entertainment [57]. Most studies have reported that enjoyment has a positive effect on purchase intention [55,56], which increases consumers' positive attitude toward specific products or services. Consumers especially experience pleasure when they purchase innovative products and support ethical brands, increasing purchase intention. Thus, GE has an effect on both consumers' attitude and behavioral intention.

Environmental concern is becoming increasingly prevalent in modern society. Laroche, Begeron, and Barbaro-Forleo [13] asserted that consumers with higher levels of environmental concern are more likely to practice environmentally friendly consumption. Therefore, when an individual perceives products and services to be environmentally friendly, their purchase intention toward such products and services is enhanced. If an individual has strong environmental ethics, they may wish to purchase green products to satisfy their hedonic goals. By using green products, consumers perceive that they have contributed to environmental conservation. GE, as the satisfaction of needs relating to the consumption of green products and services, affects consumers' GPI. Therefore, this study proposed the following hypothesis:

**Hypothesis 5 (H5).** *GE Positively Influences GPI.*

### 2.6. Positive Effect of GBL on GPI

Brand love is associated with passion-driven purchase behaviors [40], commitment, affection, connection [58], consumer–brand identification [59], and the consumer–brand relationship [60]. Batra et al. [33] argued that brand love has a significant effect on purchase intention in that it can increase consumers' willingness to pay a higher price and can generate customer value. Scholars [45,61–63] have observed that consumers frequently make purchase decisions on the basis of brand love. The present study speculated that a person with GBL toward a particular brand is more likely to specifically purchase that brand's products or services. When consumers purchase environmentally friendly or sustainable products and services, they perceive that they are contributing to the development of a circular economy. Bagozzi, Batra, and Ahuvia [11] maintained that brand love is a crucial topic among brands and marketers. Consumers' GBL increases their willingness to purchase green products. For example, BMW claims that no two vehicles it manufactures are identical. In this manner, BMW promotes a unique type of brand love for BMW based in its products' uniqueness, which can strengthen the bond BMW has with its consumers.

Green brands are brands that consumers associate with environmental conservation and eco-friendliness and that typically minimize their environmental impact in the production and shipping stages. Salehzadeh et al. [18] maintained that green brands generate benefits such as enhanced reputation through the reduction of their environmental impact. GBL is at the core of the relationship between consumers and green brands; eco-conscious consumers having a positive perception of such brands can lead to GPI and brand loyalty. Papista and Dimitriadis [64] conducted a study of brand love and its outcome variables; they indicated that the consumer–green brand relationship quality and consumer satisfaction with the green brand have a significant impact on the following behavioral outcomes: word-of-mouth (WOM), expectation of continuity, and cross-buying. Thus, they argued that green brands could be considered a means through which consumers express their environmental concerns while still enjoying similar levels of functional performance to those afforded by conventional brands. GBL increases consumers' expectancy of product quality and environmental friendliness. Therefore, GBL drives consumers to purchase eco-friendly products with a low environmental impact. GBL thus represents a strong bond between consumers and a green brand and is expected to positively affect GPI. Accordingly, this study proposed the following hypothesis:

**Hypothesis 6 (H6).** *GBL positively influences GPI.*

## 3. Methods

This study used SPSS version 22 (IBM, Armonk, NY, USA) for factor analysis. This study used structural equation modeling (SEM) to test the hypotheses. The research framework is illustrated in Figure 1.

### 3.1. Measurement Scales

This study applied two pretests involving questions on GIM and GE; after the pretest, the questionnaire was revised to ensure content validity. First, the developed questionnaire was presented during interviews with 10 part-time MBA students with a minimum of 3 years of business experience. They provided feedback about whether the questionnaire items were worded ambiguously. Additionally, the questionnaire was distributed over the Internet by using the PTT Bulletin Board System (the largest terminal-based bulletin board system based in Taiwan) to 10 consumers with a minimum of 3 years of green consumption experience. The items were scored on a 7-point Likert scale with anchors ranging from 1 ("strongly disagree") to 7 ("strongly agree").

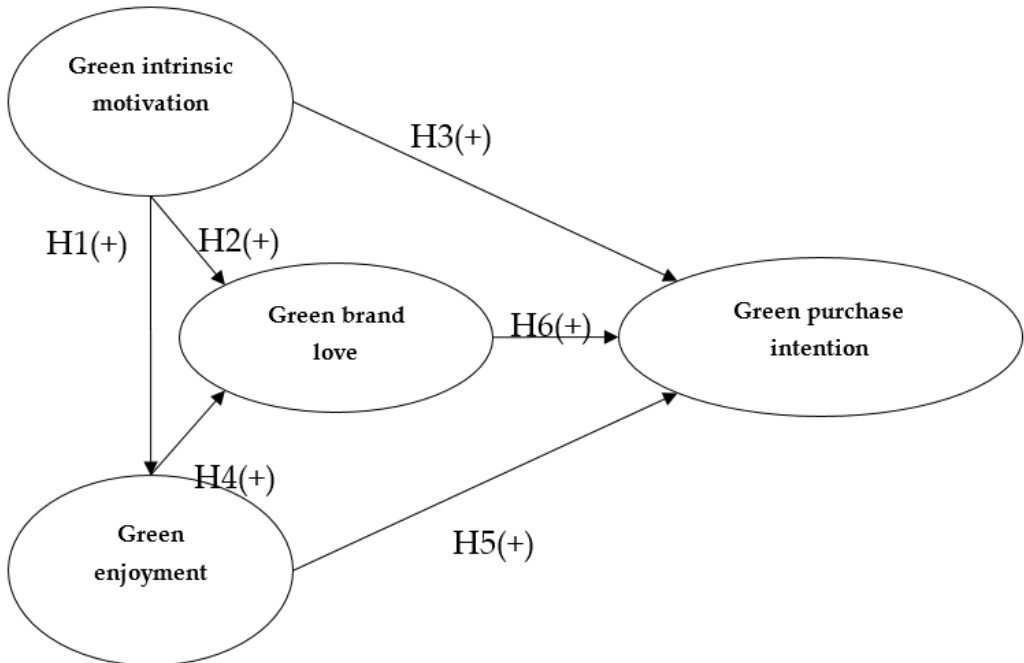

**Figure 1.** Research framework.

Tables 1 and 2 present the results of this study and other related studies. As summarized in the tables, the concepts proposed in this study are innovative and do not overlap with those in related studies. A pretest and factor analysis were conducted on the GIM, GE, GBL, and GPI subscales. According to the exploratory factor analysis results, items 1, 2, and 5 of the GIM subscale had factor loadings of 0.786, 0.860, and 0.691, respectively, and were thus removed. Items 2, 4, 5, and 7 of the GBL subscale had factor loadings of 0.864, 0.861, 0.885, and 0.891, respectively, and were thus removed. Furthermore, item 3 of the GPI subscale had a factor loading of 0.739 and was also removed. The items that had cross-loadings on more than one factor were removed to avoid multicollinearity [65]. This study identified variables relevant to GPI and used them to effectively measure GPI. The definitions and measurements of the constructs in this study are described as follows:

GIM. This study referred to Li et al. [26], who measured intrinsic motivation. To measure GIM, the following six items were used: (1) "I enjoy thinking of new green ideas", (2) "I enjoy trying to complete environmental tasks in my workplace", (3) "I enjoy tackling environmental tasks that are completely new", (4) "I enjoy improving existing green ideas in my workplace", (5) "I become excited when I have new green ideas", and (6) "I would like to become more engaged in the development of green ideas".

GE. This study referenced the survey of Tamborini et al. [32] to measure GE by applying the following five items: (1) "The products or services were enjoyable", (2) "The products or services were entertaining", (3) "The products or services were appealing", (4) "The products or services were pleasant", and (5) "The products or services were fun".

GBL. This study referred to Salehzadeh et al. [18] for the measurement of GBL, which was performed using the following seven items related to environmental products and services recently used by the respondents: (1) "This is a wonderful green brand", (2) "This green brand makes me feel good", (3) "This green brand is amazing", (4) "This green brand makes me very happy", (5) "I love this green brand", (6) "I am passionate about this green brand", and (7) "I am very attached to this green brand".

GPI. This study referred to Chen and Deng [49] to measure GPI and applied the following four items: (1) "Purchasing green products is more beneficial than purchasing nongreen products", (2) "Purchasing green energy-saving products makes me happy", (3) "When purchasing a product, I consider how it affects the environment", and (4) "I am willing to spend a little more money to purchase green products".

**Table 1.** Constructs.

| Constructs | Green Intrinsic Motivation | Green Enjoyment | Green Brand Love | Green Purchase Intention |
|---|---|---|---|---|
| Ali, Ashfaq, Begum, and Ali (2020) [34] | X | | | X |
| Li, Bhutto, Wang, Maitlo, Zafar, and Bhutto (2020) [26] | X | | | |
| Wu and Chen (2019) [17] | | | X | |
| Salehzadeh, Sayedan, Mirmehdi, and Aqagoli (2021) [18] | | | X | |
| Our paper | X | X | X | X |
| Remarks | | New concept | | |

**Table 2.** Variables.

| Constructs | Green Intrinsic Motivation | Green Enjoyment | Green Brand Love | Green Purchase Intention | Other |
|---|---|---|---|---|---|
| Ali, Ashfaq, Begum, and Ali (2020) [34] | X | | | X | Green thinking and green altruism |
| Li, Bhutto, Wang, Maitlo, Zafar, and Bhutto (2020) [26] | X | | | | Green transformational leadership, green extrinsic motivation, and green creativity |
| Wu and Chen (2019) [17] | | | X | | Green brand empowerment, green brand perceptual evaluation, green brand co-creation, green brand regret, green brand experiential satisfaction, and green brand supportive intentions |
| Salehzadeh, Sayedan, Mirmehdi, and Aqagoli (2021) [18] | | | X | | Green brand image, green trust, and green attitude |

*3.2. Data Collection and Study Sample*

This study focused on green consumers in Taiwan, which was appropriate for several reasons. Taiwan has a flourishing business-to-business and business-to-consumer e-commerce environment, the growth of which has changed the traditional business mode of in-store purchasing and generated considerable profit. Second, Taiwanese firms have developed green products and services to satisfy green customers' needs, with many of such firms using a circular economy design for sustainable environmental, engineering, and social solutions. Third, even under the ongoing economic crisis, Taiwanese consumers' attitudes and behaviors toward green products have been changing favorably, with many embracing green products and services [66]. Furthermore, Taiwan's government has committed to reducing Taiwan's dependence on imported energy resources and raw materials, thus supporting the development of a circular economy.

The questionnaire survey was distributed online through social media channels for data collection. Only people who reported having green consumption experience were eligible for inclusion in the study. E-mail addresses were collected to facilitate contact and were reviewed to ensure that they matched a standard format. To avoid common method variance, respondents were assured of their anonymity and the confidentiality of their responses, and they were requested to respond honestly to reduce the chance of social desirability bias affecting the results. The respondents were divided into two broad groups (Groups A and B) for questionnaire completion. The respondents in Group A were requested to respond to questionnaire items related to GIM and GE (Questionnaire A); concurrently, those in Group B were requested to respond to questionnaire items related to GBL and GPI (Questionnaire B). When the respondents in Group A completed the items, the author sent them the questionnaire items related to GBL and GPI (Questionnaire B); similarly, when the respondents in Group B completed the items, the author sent them the

questionnaire items related to GIM and GE (Questionnaire A). A total of 302 completed questionnaires were retrieved from 386 distributed questionnaires; the response rate was 78.24%. This study adopted convenience sampling to collect the data samples. Convenience sampling enables the researcher to freely choose sample group members. Nonprobability sampling lacks the advantage of every data sample of a particular size having an equal chance of being selected, but this is often unavoidable. Because online data collection is quick and cost-effective, many researchers have embraced this nonrepresentative method. Harvard's Project Implicit, which offers implicit-association tests, is one example. The nonprobability sampling procedure relies on the person conducting the sampling, which can elicit other complicated concerns. This person conducting the sampling must be knowledgeable of the population and phenomena being studied [67]. However, simple random sampling is a favored method for achieving sufficient external validity. Moreover, a larger sample size is required to eliminate the margin of error.

## 4. Results

All measures used were first refined using Cronbach's $\alpha$ and then tested through confirmatory factor analysis (CFA). LISREL 8.7 was applied to test the hypothesized links and the obtained results. The study employed SEM to assess both the measurement and structural models for construct validity and path analysis.

### 4.1. Measurement Model Results

Analysis of the squared multiple correlations revealed that most survey items reached the conventionally accepted threshold of 0.30 [68]. Exploratory factor analysis indicated a single-factor solution for each of the four constructs. The single-factor solutions cumulatively explained more than 60% of the variance, reducing the likelihood of common method bias. Finally, CFA was used to test the construct validity of the scales used in the study.

### 4.2. SEM Results

This study used descriptive statistics to describe the basic features of the study data. Table 3 presents the correlations between the constructs and indicates that the means and standard deviations followed normal distributions. Positive correlations were also observed among GIM, GE, GBL, and GPI (Table 3). Table 4 presents a summary of the results of the factor analyses of the data set. Every construct had only a single underlying dimension. In the present study, after the design of the questionnaire items and assessment of the factor structure according to common empirical approaches, two pretests were conducted; the test results constituted the basis for subsequent questionnaire revision to ensure the content validity. The questionnaires were distributed through online social media networks. Reliability testing was performed as follows, with the results demonstrating the validity of the measurements. First, this study tested each of the items to determine if they had significant loadings. Second, reliability was assessed using the loadings of individual items for all four constructs. After the executed exploratory factor analysis, the author removed three out of six items in the GIM dimension, four out of seven items in the GBL dimension, and one out of four items in the GPI dimension. Table 5 lists the factor loadings of all items for each construct. Cronbach's $\alpha$ was then used to calculate the internal consistency coefficients of the items in order to estimate the reliability of the questionnaire measurements. The Cronbach's $\alpha$ coefficient of GIM was 0.91, that of GE was 0.86, that of GBL was 0.62, and that of GPI was 0.89. Because the Cronbach's $\alpha$ coefficients of all four constructs exceeded 0.6 [69], the measurements were both stable and reliable. The variance inflation factor values of the exogenous constructs were all less than 5 and are listed in Table 4. This study had no multicollinearity problem [68], and the results exhibited acceptable reliability and validity.

**Table 3.** Means, Standard Deviations, and Correlation Coefficients.

| Constructs | Mean | Standard Deviation | A. | B. | C. |
|---|---|---|---|---|---|
| GIM | 4.8808 | 0.99192 | | | |
| GE | 5.1815 | 0.74846 | 0.534 ** | | |
| GBL | 4.9901 | 0.93785 | 0.568 ** | 0.637 ** | |
| GPI | 5.3035 | 0.90088 | 0.513 ** | 0.600 ** | 0.607 ** |

Note: ** $p < 0.01$.

**Table 4.** Factor Analysis, Multicollinearity Tolerance Test, and Variance Inflation Factor (VIF) Results.

| Constructs | Number of Items | Number of Factors | Accumulation on Percentage of Explained Variance | VIF | Tolerance |
|---|---|---|---|---|---|
| GIM | 3 | 1 | 71.105% | 1.647 | 0.607 |
| GE | 5 | 1 | 65.813% | 1.998 | 0.501 |
| GBL | 3 | 1 | 70.083% | 2.099 | 0.476 |
| GPI | 4 | 1 | 73.622% | 1.861 | 0.537 |

**Table 5.** Item Factor Loadings and Construct Cronbach's α Coefficients and AVE Values.

| Constructs | Items | Squared Multiple Correlation | λ | Cronbach's α | AVE | The Square Root of AVE |
|---|---|---|---|---|---|---|
| GIM | GIM1 | 0.68 | 0.78 ** | | | |
| | GIM2 | 0.72 | 0.78 ** | 0.91 | 0.58 | 0.762 |
| | GIM3 | 0.74 | 0.77 ** | | | |
| GE | GE1 | 0.66 | 0.44 ** | | | |
| | GE2 | 0.62 | 0.38 ** | | | |
| | GE3 | 0.59 | 0.35 ** | 0.86 | 0.67 | 0.819 |
| | GE4 | 0.68 | 046 ** | | | |
| | GE5 | 0.73 | 053 ** | | | |
| GAC | GAC1 | 0.71 | 0.50 ** | | | |
| | GAC2 | 0.70 | 0.79 ** | 0.62 | 0.62 | 0.787 |
| | GAC3 | 0.79 | 0.62 ** | | | |
| GCA | GPI1 | 0.71 | 0.50 ** | | | |
| | GPI2 | 0.66 | 0.44 ** | 0.89 | 0.54 | 0.735 |
| | GPI3 | 0.58 | 0.34 ** | | | |

Note: ** $p < 0.01$.

This study analyzed the validity of the four constructs through CFA. The average variance extracted (AVE), which is a measure of the variance captured by the construct in relation to the variance resulting from measurement error, was also applied to assess discriminant validity [70]. For discriminant validity, the square root of the AVE of each construct must exceed the coefficient of the correlation of this construct with any other construct. As indicated in Table 3, the model had desirable psychometric properties. For example, the square roots of the AVE for the two constructs GIM and GE were 0.762 and 0.819, respectively, and thus exceeded the correlation between the two constructs (0.534). The AVE for each latent construct exceeded the 0.5 threshold, and construct reliability must exceed 0.7 for all model constructs. Table 5 presents the factor loading, AVE, and construct reliability results. The AVE values for GIM, GE, GBL, and GPI were 0.58, 0.67, 0.62, and 0.54, respectively, all of which exceeded 0.5, thus indicating acceptable convergent validity. Model validity was determined using convergent and discriminant validity. The results revealed that the model was adaptable and suitable for assessment, and it had adequate reliability and validity. This study used the PROCESS macro version 2.15 to test the mediating effect of GE on the relationship between GIM and GBL; the effect sizes were 0.059 and 0.250, respectively. Harman's single-factor test was used to avoid common method bias. The variance value in this test was 40.94%, which was less than the threshold of 50%. Therefore, this study had no common method bias problem.

In the explicit statistical test of the measurement model and structural model, the chi-square difference test revealed that $\chi^2$ (11.9, 3) > 11.34 at the 0.01% significance level. Thus, this study assessed the hypothesized paths in the structural model. As presented in Table 6, the path analysis revealed that GIM influenced GE ($t$ = 9.15), thus supporting H1. GIM had a positive direct effect on GBL ($t$ = 7.81) and GPI ($t$ = 3.73); thus, H2 and H3 are fully supported. This finding is consistent with that obtained by Ali et al. [34]. GIM had a positive effect on GPI. Moreover, GIM had a significant total effect on GPI when influenced by GE and GBL ($t$ = 5.35). GE positively influenced GBL ($t$ = 5.39) and GPI ($t$ = 2.60), thus supporting H4 and H5. H6, which referred to the relationship between GBL and GPI, was also supported. Therefore, GIM, GE, GBL, and GPI were positively correlated and GBL positively affected GPI.

**Table 6.** Measures of Overall Model Fit.

| Hypothesis | Measurement Model Estimates | Results |
|---|---|---|
| Absolute Fit Measures | $X^2$ Sig | No, 262.47 |
| | SRMR < 0.08 | Yes, 0.052 |
| | RMSEA < 0.1 | Yes, 0.0095 |
| | GFI > 0.80 | Yes, 0.89 |
| Incremental Fit Measures | AGFI > 0.80 | Yes, 0.84 |
| | NFI > 0.80 | Yes, 0.85 |
| | NNFI > 0.90 | Yes, 0.85 |
| | CFI > 0.90 | No, 0.88 |
| | RFI > 0.90 | No, 0.79 |
| | IFI > 0.90 | No, 0.88 |
| Parsimonious Fit Measures | PNFI > 0.50 | Yes, 0.66 |
| | PGFI > 0.50 | Yes, 0.60 |
| | $\chi^2/df < 5$ | Yes, 3.70 |
| | CN > 200 | No, 110.06 |

The CFA results also indicated that the overall fit was acceptable and that the measurement model indicators were substantial and highly significant (GFI = 0.89, RMSEA = 0.095, NFI = 0.85, and CFI = 0.88). Seyal, Rahman, and Rahim [71] suggested that a GFI that exceeds 0.8 implies a satisfactory fit. Furthermore, $\chi^2/df$ must not exceed the threshold of 3 [72], and NFI must exceed the recommended value of 0.8 [73]. Moreover, the AGFI value must be higher than the recommended value of 0.8 [71]. An RMSEA value of <0.1 is desirable, but an RMSEA value of <0.08 is preferable [74]. On the basis of these criteria, the results of this study (Table 6) indicated a favorable goodness of fit; hence, the study model was determined to be reasonably consistent with the data. Table 7 lists the direct, indirect, and total effects of the factors from one SEM example.

**Table 7.** Effects of Factors Based on the Structural Equation Modeling Example.

| Path | | Coefficients | |
|---|---|---|---|
| | | Effect | *t*-Value |
| GIM → GE | | | |
| Hypothesis 1 | Direct Effect | 0.74 | 9.15 * |
| | Indirect Effect | – | – |
| | Total Effect | 0.74 | 9.15 * |
| GIM → GBL | | | |
| Hypothesis 2 | Direct Effect | 0.45 | 7.81 * |
| | Indirect Effect | 0.36 | 7.56 * |
| | Total Effect | 0.68 | 11.42 * |

**Table 7.** *Cont.*

| Path | | | Coefficients | |
|---|---|---|---|---|
| | | | Effect | *t*-Value |
| | GIM → GPI | | | |
| Hypothesis 3 | | Direct Effect | 0.37 | 3.73 * |
| | | Indirect Effect | 0.43 | 5.35 * |
| | | Total Effect | 0.80 | 13.62 * |
| | GE → GBL | | | |
| Hypothesis 4 | | Direct Effect | 0.58 | 5.39 * |
| | | Indirect Effect | – | – |
| | | Total Effect | 0.58 | 5.39 * |
| | GE → GPI | | | |
| Hypothesis 5 | | Direct Effect | 0.41 | 2.60 * |
| | | Indirect Effect | 0.25 | 2.19 * |
| | | Total Effect | 0.66 | 5.41 * |
| | GBL → GPI | | | |
| Hypothesis 6 | | Direct Effect | 0.43 | 2.28 * |
| | | Indirect Effect | – | – |
| | | Total Effect | 0.43 | 2.28 * |

Note: * $p < 0.5$.

Figure 2 depicts the study model. All six paths estimated were significant. The original figure was exported from the statistical software and is presented in Appendix A. GE positively influenced GBL and GPI. Additionally, this study revealed that GE exerted partial mediating effects on the positive relationship between GIM and GBL (0.36, * $p < 0.05$). Therefore, all study hypotheses were supported.

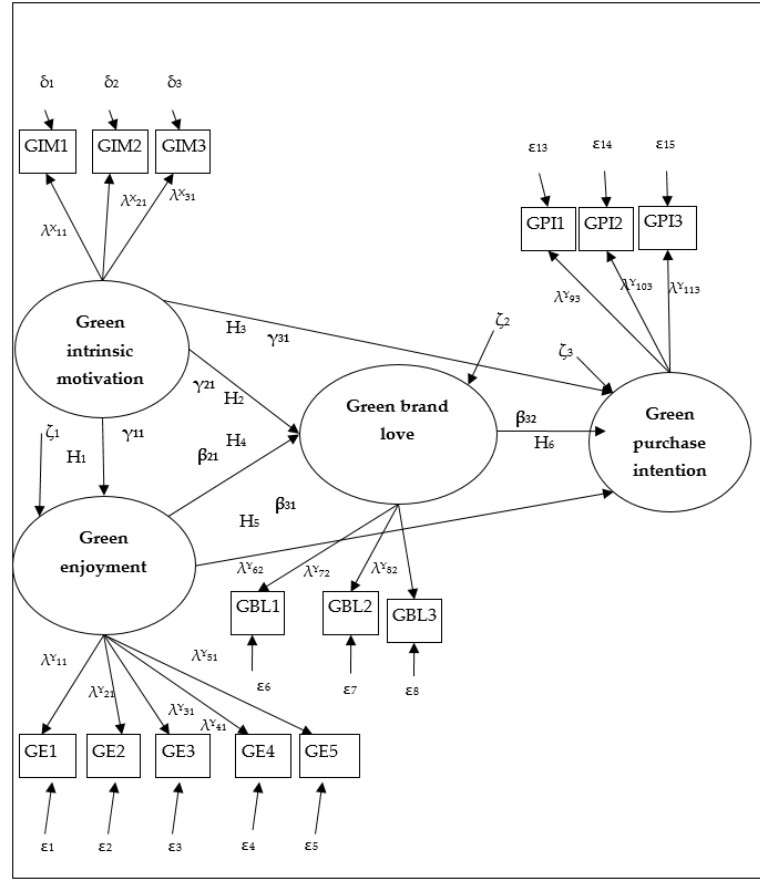

**Figure 2.** Results of the full model.

## 5. Conclusions

### 5.1. Practical Implications

The findings relating to GIM provide market managers with new insights into consumers' environmental and sustainability concerns and contribute to the current understanding of the development of a green economy. Davari and Strutton [75] maintained that consumers with a more favorable attitude toward the environment are more likely to have the intrinsic motivation to search for and use green products. Deci and Ryan [76] observed that feelings of personal expressiveness are closely related to feelings associated with intrinsic motivation. GIM is a key factor driving individuals' green purchase behaviors. Moreover, this study verified that higher levels of GIM are associated with higher levels of green consumption.

Waterman [77] indicated that when an individual is intrinsically motivated or is in a psychological state of flow, they are most likely to engage in personal expressiveness. Deci and Ryan [27] advocated that enjoyment is conceptually linked to motivation through positive feelings associated with performing an activity when intrinsically motivated. GIM is generated when people engage in an environmentally friendly activity from which they experience positive feelings. Although enjoyment is theoretically rooted in pleasure and pain, feelings of personal expressiveness are conceptually linked with feelings associated with intrinsic motivation [77]. GE is generated alongside satisfaction with and interest in green products or services. GIM guides individuals in manifesting their environmentally conscious thinking as behaviors. Hence, GIM provides a starting point for investigating GE and other variables through additional marketing-related and environment-related research.

No previous study has explored a concept similar to GE. Accordingly, the present study investigated the relationships between GE and three related concepts. GE is generated through green consumption and represents the satisfaction of the need to demonstrate environmentally friendly behaviors. This study discussed GIM, GE, GBL, and GPI, all of which together constitute the cornerstone of green business. A thorough understanding of these factors can enrich the debates on green economy theory and environmental protection strategies.

This study identified the process that forms GBL, which, alongside GE, is a product of consumer satisfaction. GBL reflects the emotion and connection consumers' have for brands with eco-friendly products and services. The results indicate that GBL increases GPI. The concept and impact of GBL is expected to attract considerable attention in the future. The results of this study reveal that GIM positively affects GE and GBL. Moreover, GE positively influences GBL and GPI. GE and GBL also positively influence GPI. These results support the proposed hypotheses. Additionally, the results indicate that GIM not only directly and positively influences green purchasing but also indirectly affects it through the partial mediating effect of GE and GBL. These empirical findings constitute valuable contributions to the literature on green business and green consumption and highlight the value of promoting GIM.

This study demonstrated that GIM positively affects GE, GBL, and GPI and that GBL mediates the positive relationship between GE and GPI. Furthermore, the mediating role of GE was identified in the present study. First, GE can indirectly influence GPI. Second, increased GE can strengthen the positive relationship between GIM and GPI. This result implies that promoting GIM and GE is a valuable measure for firms to take in their efforts to drive GPI through GBL. Third, because no previous study has investigated the role of GBL, the present study is the first to demonstrate its mediating role in the relationship between GE and GPI. Finally, this study proposes the concept of GE and thus contributes to the development of a new approach for analyzing GIM. GBL is rarely analyzed with respect to its antecedent and outcome variables. Understanding GBL in purchase behaviors could be helpful for brand managers in creating stronger brands. This study depicts GIM and GE as drivers of GBL which is relatively new. The proposed GPI model also proves that GBL reinforces the effect of brand engagement on purchase attitude.

### 5.2. Theoretical Implication

The theoretical contributions of this study are as follows. First, the idea of increasing GIM, GE, GBL, and GPI represents a novel and innovative strategy. Second, although the concepts of GBL and GPI have been examined in other studies, the present study proposed the new constructs of GIM and GE and analyzed the relationships among GIM, GE, GBL, and GPI to fill the literature gap. Third, this study integrated the concepts of GIM, GE, GBL, and GPI into a research framework of GPI to extend green economy research. Fourth, Salehzadeh et al. [18] determined that green brand image affects green brand attitude, GBL, and green trust but failed to explore the relations between GBL and other variables. This study incorporated GIM, GE, GBL, and GPI into a new model to expand the relevant literature. This study proposed the new concept GE, which can positively contribute to the development of new managerial strategies related to green business. However, further investigation is necessary to identify new green concerns; this is because changing environmental policies and challenges necessitate new strategies and solutions.

This study also revealed a path whereby GE and GBL have mediating roles in the GIM model. First, GE mediates the positive relationship between GIM and GPI. Second, GBL mediates the positive relationship between GIM and GPI. Therefore, the antecedent of GBL is GIM, and the consequence of GBL is GPI; the partial mediators are GE and GBL. GIM can directly affect GPI, and the influences of GE and GBL are also crucial. Therefore, firms can enhance their GBL through the promotion of GIM and GE. This study also identified the mediating role of GE and GBL in the relationship between GIM and GPI, such that the strong effects of GIM on GPI occur only when GE and GBL are emphasized but are weaker when GIM is emphasized alone. Most research has focused on GIM, without attempting to explore other factors. By expressly focusing on three dimensions, the present study identified the factors that drive GPI. Integrating the new concept of GE is a key step in creating, marketing, and presenting a green brand. The introduction of GE in this study, specifically in the field of green business, has changed the interpretation of the relationship between GE and GBL. The study employed the definition of enjoyment provided by Tamborini et al. [32] together with GIM and GBL to propose the research model in which GIM and GE influence GBL and GPI, which shapes GBL and GPI as consumer responses. Therefore, the mediating role of GE in the relationship between GIM and GBL was revealed. Similar studies [17,18] have failed to explore the mediating role of GBL. Thus, this study has advanced the GBL literature by exploring its antecedent and outcome variables. The results can serve as a reference for proposing theories related to GBL and GE. This study also contributed to the literature by identifying the relationships among GIM, GE, and GBL, thereby advancing green business research.

Consumer motivation and intention are crucial in green business. In the technological era, all businesses must employ innovative methods to market their products and services, and green businesses must effectively cater to their environmentally conscious consumers. Marketing strategies that focus on enhancing GIM, GE, and GBL can generate increased revenue and improve a firm's image and reputation. If a brand can reflect their eco-friendly image in their products and services, they can increase brand resonance in a green economy.

### 5.3. Limitations and Future Study

The present study includes some limitations. First, the cross-sectional approach limits the ability to make causal inferences about the findings. Second, online survey may lead to biased data, which limits the statistical power. Future research should focus on the antecedent of GBL as well as other theoretical dimensions pertinent in a green context. Furthermore, scholars could extend this framework to include other variables that may have negative effects on GPI.

**Funding:** This research received no external funding.

**Institutional Review Board Statement:** Not applicable.

**Informed Consent Statement:** Not applicable.

**Data Availability Statement:** The data that support the findings of the study are available from the author upon reasonable request.

**Acknowledgments:** The author received no financial support for the research, authorship, or publication of this article.

**Conflicts of Interest:** The author declares no conflict of interest.

**Appendix A**

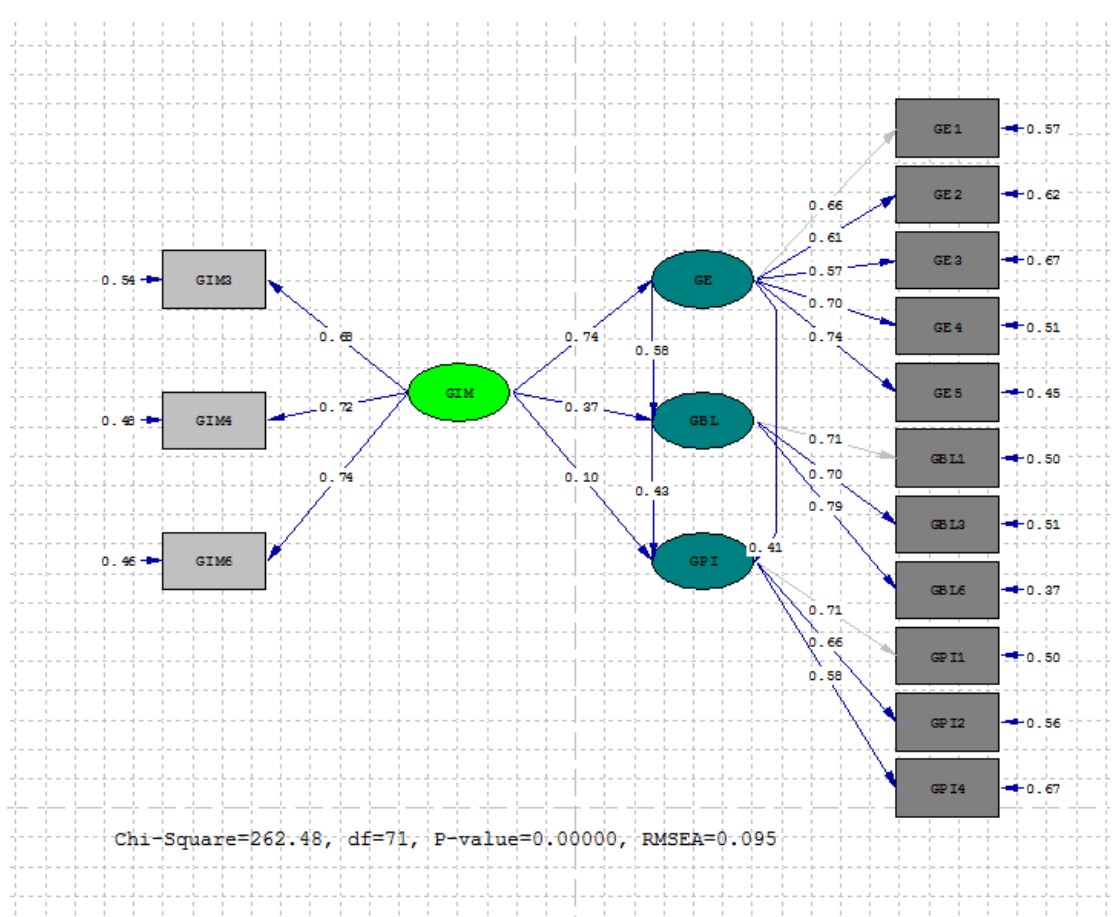

**Figure A1.** Results of the full model.

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
