# Peer review of "Determinants of Green Purchase Intention: The Roles of Green Enjoyment, Green Intrinsic Motivation, and Green Brand Love"

_sustainability, doi:10.3390/su15010132_

Round 1

Reviewer 1 Report

The research question is rather narrow and addresses a very specific sample and thus will target a niche audience.

The Title –has a different format – should be reformulated in order to expresses better the focus of the paper and not representing an enumeration of the paper keywords.

Abstract: The abstract has an atypical formulation and presents concisely the purpose and the results of the research. There are provided too many details about the methodology and the indices values, which should not be included in the abstract. There is too long description of the statistical data – which should only be checked and not detailed upon. Statistical data and figures should not be presented in detail in the Abstract. Abstract MUST be reformulated.

KEYWORDS – are accordingly, but could be improved in order to enlarge the target audience and to include the paper in a specific field.

The Introduction: tries to argue the paper scope and tries to convince about the importance that the green purchase field has but is not well enough documented. Even if is a section of subjective contribution, the Introduction should also be more documented.

Introduction presents a series of facts and events from different parts of the wolrd relates to eco-friendly behavior, but could be improved with references to other similar researches in the academia:  “The university role in developing the human capital for a sustainable bioeconomy” (https://www.econstor.eu/bitstream/10419/196452/1/Article_2743.pdf) (DOI: 10.24818/EA/2018/49/583 ) (https://doi.org/10.1080/1331677X.2022.2086597) (https://doi.org/10.3390/su132212927).   

Literature Review – Context

Literature review does not exist.

Literature review should be formulated as there are many perspectives to be debated about the core concepts the paper is built on.

Literature review should include references for supporting the researched concepts according to this journal standards.  

Literature review should refer to defining the concepts, to characterizing the concepts, to framing the concepts in the actual literature landscape.

Materials and methods / Research design

This section is atypically structured and presented.

Research methodology is presented in detail, and clearly enough in order to understand the research approach and implementation.

Data analysis stages are presented accordingly.

Hypotheses have standard formulation.

In my opinion, constructs are not clearly explained or not well- enough adapted to the investigated issue … there seems to be a bias …

Constructs are measuring very similar concepts based on very similar statements (questionnaire items) …

The definitions are not well-enough delimited / differentiated …

 I enjoy thinking of new green ideas = This green brand makes me very happy

The definitions and measurements of the constructs are detailed as follows: 309

GIM. This study referred to Li et al. [16], who measured the intrinsic motivation. To measure GIM, 310 the following six items were used: (1) “I enjoy thinking of new green ideas,” (2) “I enjoy trying to 311 complete environmental tasks in my workplace,” (3) “I enjoy tackling environmental tasks that are 312 completely new,” (4) “I enjoy improving existing green ideas in my workplace,” (5) I become excited 313 when I have new green ideas,” and (6) “I would like to become more engaged in the development of 314 green ideas.” 315

GE. This study referenced the survey of Tamborini et al. [23] to measure GE, applying the following 316 five items: (1) “The products or services were enjoyable,” (2) “The products or services were 317 entertaining,” (3) “The products or services were appealing,” (4) “The products or services were 318 pleasant,” and (5) “The products or services were fun.” 319

GBL. This study referred to Salehzadeh et al. [36] for the measurement of GBL, which was performed 320 using the following seven items regarding environmental products and services recently used by the 321 respondents: (1) “This is a wonderful green brand,” (2) “This green brand makes me feel good,” (3) 322 “This green brand is amazing,” (4) “This green brand makes me very happy,” (5) “I love this green 323 brand,” (6) “I am passionate about this green brand,” and (7) “I am very attached to this green brand.” 324

GPI. This study referred to Chen and Deng [40] to measure GPI and applied the following four items: 325 (1) “Purchasing green products is more beneficial than purchasing nongreen products,” (2) “Purchasing green energy-saving products makes me happy,” (3) “When purchasing a product, I 327 consider how it affects the environment,” and (4) “I am willing to spend a little more money to 328 purchase green products.”

Results

The Results section is well presented on paragraphs related to each investigated statistical tests and indexes.

Presentation of results is detailed and argued from statistical point of view. Results provide the validation of sample data.

Figure 2 should be exported from the statistical software, not built-in with other extensions.

Throughout the results sections there is no interpretation of the data in correlation to the investigated topic and objective. Some brief references should be introduced in order to understand the relevance, the significance of each statistical result to the general objective of the paper.

Results are not compared to previous research data.

Discussions

This section is oriented towards the presentation of quantitative results and their significance. We suggest a more detailed discussion of results from theoretical perspective also. We suggest the authors to focus more on the original contributions brought to the existing literature on the topic, to emphasize the usefulness of results for the academia and for practice.

This section should integrate arguments for supporting the research results in accordance to previous studies or research papers. The discussion section must correlate the proposed objectives of the research to the real results and provide a detailed argumentation in order to reveal the utility of the present research.

Conclusions

Conclusions are formulated rather focusing on the methodological aspects of the research. Authors insist on the correlations they assume between the proposed variables.

Conclusions should convince about the usefulness of the research and anchor its results into practical discoveries for the academia, business environment, society and other parties.

References

References are rather heterogeneous and don’t emphasize any fundamental titles or names in the domain. Could be improved mainly for the Literature review section but also within Results section.

*** English proofreading, sentence and text proofing are needed. 

Author Response

Response to Reviewer 1 Comments

Point 1: The Title –has a different format – should be reformulated in order to expresses better the focus of the paper and not representing an enumeration of the paper keywords.

Response 1: Thanks for the reviewer’s comment. In the title section, the authors revised the title, and list as follows. Determinants of Green Purchase Intention: The Nexus of Green Enjoyment, Green Intrinsic Motivation, and Green Brand Love

Point 2: Abstract: The abstract has an atypical formulation and presents concisely the purpose and the results of the research. There are provided too many details about the methodology and the indices values, which should not be included in the abstract. There is too long description of the statistical data – which should only be checked and not detailed upon. Statistical data and figures should not be presented in detail in the Abstract. Abstract MUST be reformulated.

Response 2: Thanks for the reviewer’s comment. In the abstract section, the authors revised the abstract, and list as follows. This study investigated the relationship among green enjoyment, green brand love, green intrinsic motivation, and green purchase intention. Data were collected from August 26 to September 16, 2022, through a questionnaire survey distributed online, and quantitative instruments were applied to analyze the data. A total of 302 randomly selected samples from consumers with experience of green consumption were analyzed. The data were analyzed using descriptive statistics and confirmatory factor analysis. The results revealed that the content, discriminant, and convergent validity and reliability of the model were satisfactory. Global model analysis of green intrinsic motivation revealed acceptable results. Moreover, structural equation modeling indicated a satisfactory model fit to the standard sample data. Finally, the study revealed that green intrinsic motivation positively influences green enjoyment, green brand love, and green purchase intention. Green enjoyment positively affects green brand love and green purchase intention. Furthermore, green enjoyment and green brand love mediate the positive relationship between green intrinsic motivation and green purchase intention.

Point 3: The Introduction: tries to argue the paper scope and tries to convince about the importance that the green purchase field has but is not well enough documented. Even if is a section of subjective contribution, the Introduction should also be more documented.

Introduction presents a series of facts and events from different parts of the wolrd relates to eco-friendly behavior, but could be improved with references to other similar researches in the academia:  “The university role in developing the human capital for a sustainable bioeconomy” (https://www.econstor.eu/bitstream/10419/196452/1/Article_2743.pdf) (DOI: 10.24818/EA/2018/49/583 ) (https://doi.org/10.1080/1331677X.2022.2086597) (https://doi.org/10.3390/su132212927).

Response 3: Thanks for the reviewer’s comment. In the introduction section, the author adds the explanation in introduction section, and list as follows. Please refer to the revised manuscript in the introduction section. Therefore, it would make contribute to environmental management literature and environmentally friendly industry if the terms of enjoyment became the term “green enjoyment”. Davis, Bagozzi, and Warshaw (1992) maintained that enjoyment is a pleasant feeling, and what is considered “fun” to participants. Thus, feeling pleasure about products or services in an environmentally friendly context seems to be a good idea.

Thanks for the reviewer’s comment. In the introduction section, the author adds the explanation in introduction section, and list as follows. Please refer to the revised manuscript in the introduction section. Is intrinsic motivation related to GE? Does the framework remain true if the terms of 'green' for intrinsic motivation and GE are put together in the entire framework. How the respondents knew this was green intrinsic motivation not green external motivation? At last,

Thanks for the reviewer’s comment. In the introduction section, the author adds the explanation in introduction section, and list as follows. Please refer to the revised manuscript in the introduction section. Green brand love is an increasingly important issue. With a trend of building long term relationships with consumers, brand love exists in many different industries. Big companies such as American Eagle Outfitters, Aeropostale, Express, J. Crew, and H&M are contemplating entering the market of brand love Bagozzi, Batra, and Auhvia (2017). Moreover, increasing numbers of companies are selling products and services labeled as environmentally friendly, leading to an emerging trend. In 2009, in the Grocery Manufacturers Association (GMA)/Deloitte study, GMA noted a major opportunity for companies to supply latent demands for green products. Socially-responsible or “green” goods and services are increasingly important for retailers, and their market presence continues to grow rapidly (Tully & Winer, 2014). Accompanied by appropriate regulation, trade can assist the transition to a green economy by encouraging the exchange of environmentally friendly goods and services. Tully and Winer (2014) also observed that over half of all participants in their study (60.1%) were willing to pay a premium for such products. Laroche, Michel, Bergeron, Jasmin, and Barbaro-Forleo Guido (2001) pointed out that 80% of consumers inclined to pay a premium for green products refused to purchase from companies that do not follow environmental regulations, or that misrepresent non-green products as green. Green consumption, preservation of nature and prevention of hazards typically announce green consumption simultaneously with the start of environmental protection, to both entertain existing customers and attract new ones. In terms of personal values, active participants in various environmental protection activities derive satisfaction from living frugally and engaging in environmental protection. Consequently, consumers are willing to purchase green products or services for environmental awareness and personal values. Consumers thus tend to purchase green products in the environmental era (Makower, 2009; Kalafatis, East, Pollard, & Tsogas, 1999). Some companies such as Tesla, Apple, IKEA and Johnson & Johnson already had products made out of recycled materials and aimed for the green brand love actively. With the rapidly expanding literature of brand love, researchers want to know the issue of brand love and to know further about the impact of green brand love on green purchase intention.

Point 4: Literature review: Literature review does not exist.

Literature review should be formulated as there are many perspectives to be debated about the core concepts the paper is built on.

Literature review should include references for supporting the researched concepts according to this journal standards.  

Literature review should refer to defining the concepts, to characterizing the concepts, to framing the concepts in the actual literature landscape.

Response 4: Thanks for the reviewer’s comment. This study revised the Materials and Methods section and replaced it with Literature Review and Hypothesis Development. This study uses Table 1 and 2 to position our paper. This study adds the explanation to justify why our framework is different from the existing research. Some variables that this study has founded to have much influence on green concept, are discussed in the literature review and hypothesis development section. The author adds sentences in 2.2. Positive effect of GIM on GBL section and list as follows. Please refer to the revised manuscript in the literature review and hypothesis development section. Intrinsic motivation is a person who join an activity because of its inherent satisfaction. Li et al. (2020) pointed out that green intrinsic motivation is an employee's internally driven interest, inherent love, passion, enjoyment, or satisfaction for pro-environmental behavior. Ali, Ashfaq, Begum, and Ali (2020) conducted a study in three China cities to investigate young China buyers’ behaviors toward buying electronics products. They found that green intrinsic motivation has a positive effect on green purchase intention. However, there is a lack of evidence whether green intrinsic motivation affects green brand love. This study explored the relationship between these two constructs to fulfill the research gap.

Thanks for the reviewer’s comment. This study adds the explanation to justify why our framework is different from the existing research. The author adds sentences in 2.4. Positive effect of GE on GBL section and list as follows. Please refer to the revised manuscript in the literature review and hypothesis development section. For examples, ford offers mass customization options for customers online that allows consumers have fun while choosing products. To reduce cost and increase customer responsiveness, firms often look for goods which share similar green enjoyment culture and are perceived as environmentally friendly. Green consumers tend to develop strong relationships with trustworthy brands that provide green products or services. Understanding how to initiate, develop, and maintain high quality buyer-seller relationships is critical to business success (Styles and Ambler, 1994).

Thanks for the reviewer’s comment. This study adds the explanation to justify why our framework is different from the existing research. The author adds sentences in 2.5. Positive effect of GE on GPI section and list as follows. Enjoyment is a satisfaction of needs comprised of physical, intellectual, or socially based needs (Waterman, 1993). Green enjoyment is a form of happiness that generates the satisfaction of hedonic with green products or services. Shopping is not only obtaining products or services, but also about experience, enjoyment and entertainment (Chirwa and Odhiambo, 2017). Most studies have found enjoyment has a positive effect on purchase intention (Xu et al, 2020; Wang et al 20,13) which increases consumers’ positive attitude towards specific products or services. Consumers feel pleasure especially when they buy innovative products and support ethical brands. Purchase intention generates when consumers experience enjoyment. Thus, green enjoyment has an effect on both attitude and consumers’ behavioral intention.

Thanks for the reviewer’s comment. This study adds the explanation to justify why our framework is different from the existing research. The author adds sentences in 2.6. Positive effect of GBL on GPI section and list as follows. Bagozzi, Batra, and Ahuvia (2017) maintained that brand love is an important aspect of modern brands for both consumers and marketers. Green brand love comprised of consumers’ love for and attachment to a brand led to improve their willing to buy green products. For example, BMW claims no two of its vehicles are identical. The brand love of BMW is so unique and BMW shapes a special emotion between consumers and BMW. This emotion or cognition allows consumers in the mood for green brand love to improve their green purchase intention.

Point 5: Materials and methods / Research design

This section is a typically structured and presented.

Research methodology is presented in detail, and clearly enough in order to understand the research approach and implementation.

Data analysis stages are presented accordingly.

Hypotheses have standard formulation.

In my opinion, constructs are not clearly explained or not well- enough adapted to the investigated issue … there seems to be a bias …

Constructs are measuring very similar concepts based on very similar statements (questionnaire items) …

The definitions are not well-enough delimited / differentiated …

 I enjoy thinking of new green ideas = This green brand makes me very happy

Response 5: Thanks for the reviewer’s comment. In the pretest, this study removed item 1 from GIM subscale and item 4 from GBL subscale. Therefore, there is no inadequate items in this survey.

Point 6: Results

The Results section is well presented on paragraphs related to each investigated statistical tests and indexes.

Presentation of results is detailed and argued from statistical point of view. Results provide the validation of sample data.

Figure 2 should be exported from the statistical software, not built-in with other extensions.

Throughout the results sections there is no interpretation of the data in correlation to the investigated topic and objective. Some brief references should be introduced in order to understand the relevance, the significance of each statistical result to the general objective of the paper.

Results are not compared to previous research data.

Response 6: Thanks for the reviewer’s comment. The author adds the original figure of figure 2. The original figure was exported from the statistical software and was listed in Appendix.

Thanks for the reviewer’s comment. The author revises manuscript in result section, and list as follows. This study use table 1 and 2 to present the results of this study and precious studies reported in the literature. Thus, these concepts in this study are innovative and do not overlap previous studies. This study conducted a pretest and used a factor analysis on GIM, GE, GBL, and GPI subscale. Using exploratory factor analysis, this study removed item 1, 2, and 5 from GIM subscale. These items were dropped as the factor loading were 0.786, 0.860, and 0.691, respectively. This study removed item 2, 4, 5, and 7 from GBL subscale. These items were dropped as the factor loading were 0.864, 0.861, 0.885 and 0.891, respectively. This study removed item 3 from GPI subscale. The item was dropped as the factor loading was 0.739. The items that have cross-loadings of items on more than one factor must be removed to avoid multicollinearity [63]. This study associates relevant variables to crucial green purchase intention to appropriate manage all the measurement need for green purchase intention. The definitions and measurements of the constructs in this study are described as follow:

Point 7: Discussion: This section is oriented towards the presentation of quantitative results and their significance. We suggest a more detailed discussion of results from theoretical perspective also. We suggest the authors to focus more on the original contributions brought to the existing literature on the topic, to emphasize the usefulness of results for the academia and for practice.

This section should integrate arguments for supporting the research results in accordance to previous studies or research papers. The discussion section must correlate the proposed objectives of the research to the real results and provide a detailed argumentation in order to reveal the utility of the present research.

Response 7: Thanks for the reviewer’s comment. The author revises manuscript in discussion section, and list as follows. In explicit statistical test of measurement model and structural model, the chi-square difference test is [Chi-square=11.9(3)> Chi-square=11.34] at 0.01 level of significance. Thus, this study conducted an assessment of the hypothesized paths in the structural model. From path analysis, GIM influenced GE (t=9.15), thus supporting H1. GIM have a positive direct effect on GBL (t=7.81) and GPI (t=3.73), thus H2 and H3 are fully supported. This this finding is consistent with Ali et al.’s (2020) result. GIM has a positive effect on GPI. Moreover, GIM had a significant total effect on GPI when intervened by GE and GBL (t=5.35). Meanwhile, GE positively influenced GBL (t=5.39) and GPI (t=2.60), thus supporting H4 and H5. H6, which tested the relationship between GBL and GPI, also was supported (see Table 6 for full results). GIM, GE, GBL, and GPI are positively correlated. GBL does positively affect GPI.

Point 8: Conclusions

Conclusions are formulated rather focusing on the methodological aspects of the research. Authors insist on the correlations they assume between the proposed variables.

Conclusions should convince about the usefulness of the research and anchor its results into practical discoveries for the academia, business environment, society and other parties.

Response 8: Thanks for the reviewer’s comment.  The author adds 5.1 Practical implication and 5.2 Theoretical implication. The author revises manuscript in conclusion section, and list as follows. Although enjoyment is theoretically based in pleasure and pain, feelings of personal expressiveness are conceptually linked with feelings associated with intrinsic motivation (Deci & Ryan, 1985). Green enjoyment is a process alongside satisfaction and interest about green products or services.

The author revises manuscript in conclusion section, and list as follows. Fifth, Salehzadeh et al. (2021) found that green brand image affects green brand attitude, green brand love and green trust. But they failed to explore the relations between green brand love and other variables. This study posited GIM, GE, GBL, and GPI into a new model to leads to a genuine understanding of literary works. This study posits the new concept GE, and thus contributes to the development of new managerial theories to enrich green business.

The author revises manuscript in conclusion section, and list as follows. the partial mediators are GE and GBL. GIM can directly affect GPI and the influence of GE and GBL are important as well.

Point 9: References

References are rather heterogeneous and don’t emphasize any fundamental titles or names in the domain. Could be improved mainly for the Literature review section but also within Results section.

Response 9: Thanks for the reviewer’s comment. The author adds literature in the reference section. The author lists it as follows.

Davis, F. D., Bagozzi, R., & Warshaw, P. Extrinsic and intrinsic motivation to use computers in the workplace. J. Appl. Soc. Psychol. 1992, 22, 1111-1132.

Tully, S.M., Winer R.S. The Role of the Beneficiary in Willingness to Pay for Socially Responsible Products: A Meta-analysis. J. Retail. 2014, 90, 255-274.

Laroche, M., Bergeron, J., Barbaro-Forleo, G. Targeting consumers who are willing to pay more for environmentally friendly products. J. Consum. Mark. 2001, 18, 503-520.

Bratianu, C. Stanescu D.F., Mocanu, R., Bejinaru, R. Serial Multiple Mediation of the Impact of Customer Knowledge Management on Sustainable Product Innovation by Innovative Work Behavior. Sustainability. 2021, 13, 12927.

Makower, J. Strategies for Green Economy – Opportunities and Challenges in the New World of Business (2nd ed.), McGraw Hill Companies Inc: New York, NY, US, 2009.

Kalafatis, S., Pollard, M., East R., Tsogas M. Green marketing and Ajzen's theory of planned behavior A cross-market examination, J. Consum. Mark. 1999, 16, 441-460.

Ali, F., Ashfaq, M., Begum, S., Ali, A. How “Green” thinking and altruism translate into purchasing intentions for electronics products: The intrinsic-extrinsic motivation mechanism. Sustain. Prod. Consum. 2020, 24, 281–291.

Styles, C. Ambler, T. Successful export practice The UK experience, Int. Mark. Rev. 1994, 11, 23–47

Chirwa, T.G., Odhiambo, N.M. Sources of Economic Growth in Zambia: An Empirical Investigation.       Glob. Bus. Rev. 2017, 18, 275-290.

Chen, Y.-S., Huang, A.-F., Wang, T.-Y., Chen Y.-R. Greenwash and green purchase behavior: the mediation of green brand image and green brand loyalty, Total. Qual. Manag. Bus. Excell. 2020, 31, 194-209.

Kerlinger, F.N. Lee H.B. Foundations of Behavioral Research (4th edition), Cengage Learning: Boston, Massachusetts, US, 2000.

Reviewer 2 Report

Determinants of Green Purchase Intention: The Roles of Green Intrinsic Motivation, Green Enjoyment and Green Brand Love

This paper investigates a series of ‘green’ concepts in relation to consumer behaviour/purchasing intentions. These ‘green’ concepts vaguely represent a mix of psychological attitudes of consumers assumed to be experienced when considering or consuming a perceived sustainable/ethical product. While there are interesting principles presented in this paper, there is little explanation of the concepts and their proposed relationship that would help the reader derive any benefit from this paper. Furthermore the paper has flaws in how the study is presented, which reduces the apparent credibility of the study. Below is a brief list of the issues that would need to be resolved to reach the standard expected for publication.

The paper currently lacks a theoretical basis and is poorly structured. It is also unconvincing in demonstrating any specific theory contribution. Below are brief thoughts on the paper.

1. The conceptual labels were briefly outlined by largely leaning on other authors, but there was insufficient definition. The initial section did not provide any adequate interrogation of the underlying concepts. 

2. Once the concepts have been adequately defined, the paper needs a present a more thorough discussion and rigorous unpacking of the assumed relationships between these concepts. 

2. The paper is currently lacking in critical depth and as a result remains theoretically unconvincing. A firmer theoretical base is necessary to move this paper forward. Previous studies are listed and fleetingly described, but there is little attempt to offer an adequate evaluation. 

4. Related to the earlier points, the hypotheses needed a stronger basis from which to proceed the remainder of the study.

5. The research methods and measurement scale need a clearer description and better justification. 

6. The results of the study were not presented before the discussion section, resulting in a disjointed manuscript; present the results before the discussion section. 

7. The discussion section begins with comments relating to the methods of analysis, which should be in the research design/methods section (which goes before the results section). As such there is very little discussion of the findings.

8. Due to the limited discussion of findings, the contribution is not adequately discussed and thus remains very unclear. And without a clear contribution it is difficult to see value in the paper in its current format.

Summary: Although it pains me to be so blunt, this is unfortunately a weak paper that would provide no benefit to a general reader. This paper might have more merit if it is thoroughly revised and targeted at a more specialised journal.

Author Response

Response to Reviewer 2 Comments

This paper investigates a series of ‘green’ concepts in relation to consumer behaviour/purchasing intentions. These ‘green’ concepts vaguely represent a mix of psychological attitudes of consumers assumed to be experienced when considering or consuming a perceived sustainable/ethical product. While there are interesting principles presented in this paper, there is little explanation of the concepts and their proposed relationship that would help the reader derive any benefit from this paper. Furthermore the paper has flaws in how the study is presented, which reduces the apparent credibility of the study. Below is a brief list of the issues that would need to be resolved to reach the standard expected for publication.

The paper currently lacks a theoretical basis and is poorly structured. It is also unconvincing in demonstrating any specific theory contribution. Below are brief thoughts on the paper.

Point 1:

The conceptual labels were briefly outlined by largely leaning on other authors, but there was insufficient definition. The initial section did not provide any adequate interrogation of the underlying concepts.

Response 1: Thanks for the reviewer’s comment. In the introduction section, the author adds the explanation in introduction section, and list as follows. Please refer to the revised manuscript in the introduction section. Therefore, it would make contribute to environmental management literature and environmentally friendly industry if the terms of enjoyment became the term “green enjoyment”. Davis, Bagozzi, and Warshaw (1992) maintained that enjoyment is a pleasant feeling, and what is considered “fun” to participants. Thus, feeling pleasure about products or services in an environmentally friendly context seems to be a good idea.

Thanks for the reviewer’s comment. In the introduction section, the author adds the explanation in introduction section, and list as follows. Please refer to the revised manuscript in the introduction section. Is intrinsic motivation related to GE? Does the framework remain true if the terms of 'green' for intrinsic motivation and GE are put together in the entire framework. How the respondents knew this was green intrinsic motivation not green external motivation? At last,

Thanks for the reviewer’s comment. In the introduction section, the author adds the explanation in introduction section, and list as follows. Please refer to the revised manuscript in the introduction section. Green brand love is an increasingly important issue. With a trend of building long term relationships with consumers, brand love exists in many different industries. Big companies such as American Eagle Outfitters, Aeropostale, Express, J. Crew, and H&M are contemplating entering the market of brand love Bagozzi, Batra, and Auhvia (2017). Moreover, increasing numbers of companies are selling products and services labeled as environmentally friendly, leading to an emerging trend. In 2009, in the Grocery Manufacturers Association (GMA)/Deloitte study, GMA noted a major opportunity for companies to supply latent demands for green products. Socially-responsible or “green” goods and services are increasingly important for retailers, and their market presence continues to grow rapidly (Tully & Winer, 2014). Accompanied by appropriate regulation, trade can assist the transition to a green economy by encouraging the exchange of environmentally friendly goods and services. Tully and Winer (2014) also observed that over half of all participants in their study (60.1%) were willing to pay a premium for such products. Laroche, Michel, Bergeron, Jasmin, and Barbaro-Forleo Guido (2001) pointed out that 80% of consumers inclined to pay a premium for green products refused to purchase from companies that do not follow environmental regulations, or that misrepresent non-green products as green. Green consumption, preservation of nature and prevention of hazards typically announce green consumption simultaneously with the start of environmental protection, to both entertain existing customers and attract new ones. In terms of personal values, active participants in various environmental protection activities derive satisfaction from living frugally and engaging in environmental protection. Consequently, consumers are willing to purchase green products or services for environmental awareness and personal values. Consumers thus tend to purchase green products in the environmental era (Makower, 2009; Kalafatis, East, Pollard, & Tsogas, 1999). Some companies such as Tesla, Apple, IKEA and Johnson & Johnson already had products made out of recycled materials and aimed for the green brand love actively. With the rapidly expanding literature of brand love, researchers want to know the issue of brand love and to know further about the impact of green brand love on green purchase intention.

Point 2: Once the concepts have been adequately defined, the paper needs a present a more thorough discussion and rigorous unpacking of the assumed relationships between these concepts. 

Response 2: Thanks for the reviewer’s comment. This study revised the Materials and Methods section and replaced it with Literature Review and Hypothesis Development. This study uses Table 1 and 2 to position our paper. This study adds the explanation to justify why our framework is different from the existing research. Some variables that this study has founded to have much influence on green concept, are discussed in the literature review and hypothesis development section. The author adds sentences in 2.2. Positive effect of GIM on GBL section and list as follows. Please refer to the revised manuscript in the literature review and hypothesis development section. Intrinsic motivation is a person who join an activity because of its inherent satisfaction. Li et al. (2020) pointed out that green intrinsic motivation is an employee's internally driven interest, inherent love, passion, enjoyment, or satisfaction for pro-environmental behavior. Ali, Ashfaq, Begum, and Ali (2020) conducted a study in three China cities to investigate young China buyers’ behaviors toward buying electronics products. They found that green intrinsic motivation has a positive effect on green purchase intention. However, there is a lack of evidence whether green intrinsic motivation affects green brand love. This study explored the relationship between these two constructs to fulfill the research gap.

Thanks for the reviewer’s comment. This study adds the explanation to justify why our framework is different from the existing research. The author adds sentences in 2.4. Positive effect of GE on GBL section and list as follows. Please refer to the revised manuscript in the literature review and hypothesis development section. For examples, ford offers mass customization options for customers online that allows consumers have fun while choosing products. To reduce cost and increase customer responsiveness, firms often look for goods which share similar green enjoyment culture and are perceived as environmentally friendly. Green consumers tend to develop strong relationships with trustworthy brands that provide green products or services. Understanding how to initiate, develop, and maintain high quality buyer-seller relationships is critical to business success (Styles and Ambler, 1994).

Thanks for the reviewer’s comment. This study adds the explanation to justify why our framework is different from the existing research. The author adds sentences in 2.5. Positive effect of GE on GPI section and list as follows. Enjoyment is a satisfaction of needs comprised of physical, intellectual, or socially based needs (Waterman, 1993). Green enjoyment is a form of happiness that generates the satisfaction of hedonic with green products or services. Shopping is not only obtaining products or services, but also about experience, enjoyment and entertainment (Chirwa and Odhiambo, 2017). Most studies have found enjoyment has a positive effect on purchase intention (Xu et al, 2020; Wang et al 20,13) which increases consumers’ positive attitude towards specific products or services. Consumers feel pleasure especially when they buy innovative products and support ethical brands. Purchase intention generates when consumers experience enjoyment. Thus, green enjoyment has an effect on both attitude and consumers’ behavioral intention.

Thanks for the reviewer’s comment. This study adds the explanation to justify why our framework is different from the existing research. The author adds sentences in 2.6. Positive effect of GBL on GPI section and list as follows. Bagozzi, Batra, and Ahuvia (2017) maintained that brand love is an important aspect of modern brands for both consumers and marketers. Green brand love comprised of consumers’ love for and attachment to a brand led to improve their willing to buy green products. For example, BMW claims no two of its vehicles are identical. The brand love of BMW is so unique and BMW shapes a special emotion between consumers and BMW. This emotion or cognition allows consumers in the mood for green brand love to improve their green purchase intention.

Point 3: The paper is currently lacking in critical depth and as a result remains theoretically unconvincing. A firmer theoretical base is necessary to move this paper forward. Previous studies are listed and fleetingly described, but there is little attempt to offer an adequate evaluation. 

Response 3: Thanks for the reviewer’s comment. In the introduction section, the author adds the explanation introduction section, and list as follows.

Green brand love is an increasingly important issue. With a trend of building long term relationships with consumers, brand love exists in many different industries. Big companies such as American Eagle Outfitters, Aeropostale, Express, J. Crew, and H&M are contemplating entering the market of brand love Bagozzi, Batra, and Auhvia (2017). Moreover, increasing numbers of companies are selling products and services labeled as environmentally friendly, leading to an emerging trend. In 2009, in the Grocery Manufacturers Association (GMA)/Deloitte study, GMA noted a major opportunity for companies to supply latent demands for green products. Socially-responsible or “green” goods and services are increasingly important for retailers, and their market presence continues to grow rapidly (Tully & Winer, 2014). Accompanied by appropriate regulation, trade can assist the transition to a green economy by encouraging the exchange of environmentally friendly goods and services. Tully and Winer (2014) also observed that over half of all participants in their study (60.1%) were willing to pay a premium for such products. Laroche, Michel, Bergeron, Jasmin, and Barbaro-Forleo Guido (2001) pointed out that 80% of consumers inclined to pay a premium for green products refused to purchase from companies that do not follow environmental regulations, or that misrepresent non-green products as green. Green consumption, preservation of nature and prevention of hazards typically announce green consumption simultaneously with the start of environmental protection, to both entertain existing customers and attract new ones. In terms of personal values, active participants in various environmental protection activities derive satisfaction from living frugally and engaging in environmental protection. Consequently, consumers are willing to purchase green products or services for environmental awareness and personal values. Consumers thus tend to purchase green products in the environmental era (Makower, 2009; Kalafatis, East, Pollard, & Tsogas, 1999). Some companies such as Tesla, Apple, IKEA and Johnson & Johnson already had products made out of recycled materials and aimed for the green brand love actively. With the rapidly expanding literature of brand love, researchers want to know the issue of brand love and to know further about the impact of green brand love on green purchase intention.

Point 4: Related to the earlier points, the hypotheses needed a stronger basis from which to proceed the remainder of the study.

.

Response 4: Thanks for the reviewer’s comment. This study revised the Materials and Methods section and replaced it with Literature Review and Hypothesis Development. This study uses Table 1 and 2 to position our paper. This study adds the explanation to justify why our framework is different from the existing research. Some variables that this study has founded to have much influence on green concept, are discussed in the literature review and hypothesis development section. The author list as follows. Please refer to the revised manuscript in the literature review and hypothesis development section. Intrinsic motivation is a person who join an activity because of its inherent satisfaction. Li et al (2020) pointed out that green intrinsic motivation is an employee's internally driven interest, inherent love, passion, enjoyment, or satisfaction for pro-environmental behavior. Ali, Ashfaq, Begum, and Ali (2020) conducted a study in three China cities to investigate young China buyers’ behaviors toward buying electronics products. They found that green intrinsic motivation has a positive effect on green purchase intention. However, there is a lack of evidence whether green intrinsic motivation affects green brand love. This study explored the relationship between these two constructs to fulfill the research gap.

For examples, ford offers mass customization options for customers online that allows consumers have fun while choosing products. To reduce cost and increase customer responsiveness, firms often look for goods which share similar green enjoyment culture and are perceived as environmentally friendly. Green consumers tend to develop strong relationships with trustworthy brands that provide green products or services. Understanding how to initiate, develop, and maintain high quality buyer-seller relationships is critical to business success (Styles and Ambler, 1994).

Point 5: The research methods and measurement scale need a clearer description and better justification. 

Response 5: Thanks for the reviewer’s comment. The author adds the original figure of figure 2. The original figure was exported from the statistical software and was listed in Appendix.

Thanks for the reviewer’s comment. The author revises manuscript in result section, and list as follows. This study use table 1 and 2 to present the results of this study and precious studies reported in the literature. Thus, these concepts in this study are innovative and do not overlap previous studies. This study conducted a pretest and used a factor analysis on GIM, GE, GBL, and GPI subscale. Using exploratory factor analysis, this study removed item 1, 2, and 5 from GIM subscale. These items were dropped as the factor loading were 0.786, 0.860, and 0.691, respectively. This study removed item 2, 4, 5, and 7 from GBL subscale. These items were dropped as the factor loading were 0.864, 0.861, 0.885 and 0.891, respectively. This study removed item 3 from GPI subscale. The item was dropped as the factor loading was 0.739. The items that have cross-loadings of items on more than one factor must be removed to avoid multicollinearity [63]. This study associates relevant variables to crucial green purchase intention to appropriate manage all the measurement need for green purchase intention. The definitions and measurements of the constructs in this study are described as follow:

Point 6: The results of the study were not presented before the discussion section, resulting in a disjointed manuscript; present the results before the discussion section.

Response 6: Thanks for the reviewer’s comment. The author adds the original figure of figure 2. The original figure was exported from the statistical software and was listed in Appendix.

Thanks for the reviewer’s comment. The author revises manuscript in result section, and list as follows. This study use table 1 and 2 to present the results of this study and precious studies reported in the literature. Thus, these concepts in this study are innovative and do not overlap previous studies. This study conducted a pretest and used a factor analysis on GIM, GE, GBL, and GPI subscale. Using exploratory factor analysis, this study removed item 1, 2, and 5 from GIM subscale. This study removed item 2, 4, 5, and 7 from GBL subscale. This study removed item 3 from GPI subscale. The definitions and measurements of the constructs in this study are described as follow. This study associates relevant variables to crucial green purchase intention to appropriate manage all the measurement need for green purchase intention.

Point 7: The discussion section begins with comments relating to the methods of analysis, which should be in the research design/methods section (which goes before the results section). As such there is very little discussion of the findings.

Response 7: Thanks for the reviewer’s comment. The author revises manuscript in discussion section, and list as follows. In explicit statistical test of measurement model and structural model, the chi-square difference test is [Chi-square=11.9(3)> Chi-square=11.34] at 0.01 level of significance. Thus, this study conducted an assessment of the hypothesized paths in the structural model. From path analysis, GIM influenced GE (t=9.15), thus supporting H1. GIM have a positive direct effect on GBL (t=7.81) and GPI (t=3.73), thus H2 and H3 are fully supported. This this finding is consistent with Ali et al.’s (2020) result. GIM has a positive effect on GPI. Moreover, GIM had a significant total effect on GPI when intervened by GE and GBL (t=5.35). Meanwhile, GE positively influenced GBL (t=5.39) and GPI (t=2.60), thus supporting H4 and H5. H6, which tested the relationship between GBL and GPI, also was supported (see Table 6 for full results). GIM, GE, GBL, and GPI are positively correlated. GBL does positively affect GPI.

Point 8: Due to the limited discussion of findings, the contribution is not adequately discussed and thus remains very unclear. And without a clear contribution it is difficult to see value in the paper in its current format.

Response 8: Thanks for the reviewer’s comment.  The author adds 5.1 Practical implication and 5.2 Theoretical implication. The author revises manuscript in conclusion section, and list as follows. Although enjoyment is theoretically based in pleasure and pain, feelings of personal expressiveness are conceptually linked with feelings associated with intrinsic motivation (Deci & Ryan, 1985). Green enjoyment is a process alongside satisfaction and interest about green products or services.

The author revises manuscript in conclusion section, and list as follows. Fifth, Salehzadeh et al. (2021) found that green brand image affects green brand attitude, green brand love and green trust. But they failed to explore the relations between green brand love and other variables. This study posited GIM, GE, GBL, and GPI into a new model to leads to a genuine understanding of literary works. This study posits the new concept GE, and thus contributes to the development of new managerial theories to enrich green business.

The author revises manuscript in conclusion section, and list as follows. the partial mediators are GE and GBL. GIM can directly affect GPI and the influence of GE and GBL are important as well.

Reviewer 3 Report

The authors investigated the relationship among green enjoyment, green brand love, green intrinsic motivation, and green purchase intention. The research approach is Structural Equation Modelling (SEM). As this kind of strategy requires, there is a strong theoretical basis for all proposed constructs and the path model.

I am very impressed with the robustness of the methodology presented by the authors. My main (but not serious) concern is about the sampling method and the limitations of the results.

Revisions:

- Even considering some redundancy, please explain the measurements of all constructs in section “3.1. Measurement Scales”;

- In “3.1.1. Data collection and the sample” please provide one or more references to endorse the statements of the first paragraph;

- Also in 3.1.1, please discuss the limitations of the sampling method. Although well planned, it seems to be a convenience sample;

-  Still in 3.1.1, the definitions and measurements of the constructs’ details  should be more adequately placed in section 3.1;

- Then renumber 3.1.1 as 3.2;

- In Table 5 please put the hypotheses symbols in the first column.

Finally, please see the attached file with minor revisions.

Author Response

Response to Reviewer 3 Comments

The authors investigated the relationship among green enjoyment, green brand love, green intrinsic motivation, and green purchase intention. The research approach is Structural Equation Modelling (SEM). As this kind of strategy requires, there is a strong theoretical basis for all proposed constructs and the path model.

I am very impressed with the robustness of the methodology presented by the authors. My main (but not serious) concern is about the sampling method and the limitations of the results.

Point 1: Even considering some redundancy, please explain the measurements of all constructs in section “3.1. Measurement Scales”;

Response 1: Thanks for the reviewer’s comment. The author revises manuscript in result section, and list as follows. This study use table 1 and 2 to present the results of this study and precious studies reported in the literature. Thus, these concepts in this study are innovative and do not overlap previous studies. This study conducted a pretest and used a factor analysis on GIM, GE, GBL, and GPI subscale. Using exploratory factor analysis, this study removed item 1, 2, and 5 from GIM subscale. This study removed item 2, 4, 5, and 7 from GBL subscale. This study removed item 3 from GPI subscale. This study associates relevant variables to crucial green purchase intention to appropriate manage all the measurement need for green purchase intention. The definitions and measurements of the constructs in this study are described as follow:

Point 2: In “3.1.1. Data collection and the sample” please provide one or more references to endorse the statements of the first paragraph;

Response 2: Thanks for the reviewer’s comment. The author adds literature in the Data collection and the sample section. The author lists it as follows. Third, even under the ongoing economic crisis, Taiwanese consumers’ attitudes and behaviors toward green products have been changing favorably, with many embracing green products and services (Chen et al., 2020).

Point 3: Also in 3.1.1, please discuss the limitations of the sampling method. Although well planned, it seems to be a convenience sample

Response 3: Thanks for the reviewer’s comment. In the introduction section, the author add the explanation in 3.2 section, and list as follows. This study adopted convenience sampling to collect data samples. Convenience sampling allows researcher free to choose sample group members according to his will. Nonprobability sampling lacks the virtues that every sample of a particular size has an equal chance of being selected but is still often necessary and unavoidable. Due to online data collection is relative cheap and fast, many researchers, have embraced these non-representative methods. Harvard’s Project Implicit, which offers Implicit Association Tests, is a good example. With nonprobability sampling the emphasis relies on the person doing the sampling, and that can bring with it an entirely new and complicated batch of concerns. This person doing the sampling must be knowledgeable of the population to be studied and the phenomena under study (Kerlinger and Lee, 2000, p.178). However, A simple random sampling is a better solution to have enough external validity. Moreover, larger sample size is needed to eliminate the margin of error.

Point 4: Still in 3.1.1, the definitions and measurements of the constructs’ details  should be more adequately placed in section 3.1;

Response 4: Thanks for the reviewer’s comment. This study moved the paragraph to the 3.1. Measurement Scales section, and list as follows. This study use table 1 and 2 to present the results of this study and precious studies reported in the literature. Thus, these concepts in this study are innovative and do not overlap previous studies. This study conducted a pretest and used a factor analysis on GIM, GE, GBL, and GPI subscale. Using exploratory factor analysis, this study removed item 1, 2, and 5 from GIM subscale. These items were dropped as the factor loading were 0.786, 0.860, and 0.691, respectively. This study removed item 2, 4, 5, and 7 from GBL subscale. These items were dropped as the factor loading were 0.864, 0.861, 0.885 and 0.891, respectively. This study removed item 3 from GPI subscale. The item was dropped as the factor loading was 0.739. The items that have cross-loadings of items on more than one factor must be removed to avoid multicollinearity [63]. This study associates relevant variables to crucial green purchase intention to appropriate manage all the measurement need for green purchase intention. The definitions and measurements of the constructs in this study are described as follow:

Point 5: Then renumber 3.1.1 as 3.2

Response 5: Thanks for the reviewer’s comment. This study revised the number 3.1.1 and replaced it with 3.2.

Point 6: In Table 5 please put the hypotheses symbols in the first column.

Response 6: Thanks for the reviewer’s comment. This study modified table 5 into table 7, and added hypotheses symbols in the first column. This study lists as follows.

Table 7. Effects of Factors Based on the Structural Equation Modeling Example

Path

Coefficients

Effect

t-Value

GIM →GE

Hypothesis 1

Direct Effect

0.74

9.15*

Indirect Effect

--

--

Total Effect

0.74

9.15*

GIM → GBL

Hypothesis 2

Direct Effect

0.45

7.81*

Indirect Effect

0.36

7.56*

Total Effect

0.68

11.42*

GIM → GPI

Hypothesis 3

Direct Effect

0.37

3.73*

Indirect Effect

0.43

5.35*

Total Effect

0.80

13.62*

GE → GBL

Hypothesis 4

Direct Effect

0.58

5.39*

Indirect Effect

--

--

Total Effect

0.58

5.39*

GE → GPI

Hypothesis 5

Direct Effect

0.41

2.60*

Indirect Effect

0.25

2.19*

Total Effect

0.66

5.41*

GBL → GPI

Hypothesis 6

Direct Effect

0.43

2.28*

Indirect Effect

--

--

Total Effect

0.43

2.28*

Note: * p<0.5

Round 2

Reviewer 2 Report

While the authors have acknowledged the limitations, the additional material does not significantly enhance the paper. The criticality of the writing and the depth of enquiry remains an issue throughout. The additional references in the introduction, used as an attempt to establish consumer appetite, are out of date (from 23 years old to 8 years old): drawing on consumer-preference data from 1999, 2001, 2009, 2014 does little to establish the relevance of the study. Also, the section titled "theoretical implications" merely lists supposed contributions, and does not discuss the "implications".

Author Response

Point 1:

While the authors have acknowledged the limitations, the additional material does not significantly enhance the paper. The criticality of the writing and the depth of enquiry remains an issue throughout. The additional references in the introduction, used as an attempt to establish consumer appetite, are out of date (from 23 years old to 8 years old): drawing on consumer-preference data from 1999, 2001, 2009, 2014 does little to establish the relevance of the study. Also, the section titled "theoretical implications" merely lists supposed contributions, and does not discuss the "implications".

Response 1: Thanks for the reviewer’s comment. In the introduction section, the author had added the explanation and reference in the earlier version. The author had added Bagozzi, Batra, and Ahuvia’s (2017) article to improve credibility when the author resubmitted the paper. This time the author further adds the explanation in introduction section, and list as follows. Please refer to the revised manuscript in the introduction section. The UK Conservative and Liberal Democrat government environmental policy for period 2020 to 2025 point out two important policies: The first, the UK government will Support innovation that make products and services more environmentally friendly. The second, the UK government will encourage resource efficiency and environmental management (PHE 2020). At the same time, the nexus of environmental economy is a prominent issue in big companies.

The author adds the explanation in introduction section, and list as follows. Please refer to the revised manuscript in the introduction section. Previous studies have utilized green brand love as the antecedent constructs (Wu and Chen, 2019) and outcome constructs (Wu and Chen, 2019; Salehzadeh, Sayedan, Mirmehdi, and Aqagoli, 2021), but have failed to explore the mediating role of green brand love.

Thanks for the reviewer’s comment. The author adds the paragraph in the conclusion section, and list as follows. Most previous research has focused on GIM, without attempting to explore other factors. By expressly focusing on three dimensions, the current study aims to identify the factors that drive GPI. Thinking about the creating, marketing and presentation of the brand is especially important when you come up with a new concept "green enjoyment". The description of GE in this article, specifically in the field of green business, has created a drastic change in the relationship between GE and GBL. The study employs Tamborini et al.'s (2010) definition of enjoyment together with GIM and GBL to propose the research model, in which GIM and GE influence GBL and GPI, and then shape GBL and GPI as consumers' responses. Therefore, the mediating role of GE in the relationship between GIM and GBL was revealed. Previous similar studies (Wu and Chen, 2019; Salehzadeh, Sayedan, Mirmehdi, and Aqagoli, 2021) failure to explore the mediating role of GBL. Thus, this study has advanced the green brand love literature in exploring the antecedent and outcome variables of GBL. The results will serve as a reference on which to base the proposed theory about GBL and GE. Besides, it is important to conduct this study because this study contributes to the literature by identifying the relationship among GIM, GE, and GBL, which further offers a valuable and significant concept for the green business literature.

This manuscript was edited by Wallace Academic Editing.

Round 3

Reviewer 2 Report

The addition of more recent data improves the manuscript marginally. However the revisions to the article, to my mind, do not adequately address the issues raised initially. "Criticality" is lacking throughout the manuscript. While the discussion section has been revised there is still an underlying weaknesses. In the example below, the first two sentences clearly show a misunderstanding of what constitutes a contribution: devising and exploring a piece of research is not a theoretical contribution.

"5.2 Theoretical implication
The theoretical contributions of this study are as follows. First, the study devised and explored a green economy research framework."

The issues previously raised still remain in the manuscript, and unfortunately there are no easy, quick fixes that can be solved by adding a few sentences or more recent references. A more detailed critical revision of the manuscript is necessary.

Author Response

Response to Reviewer 2 Comments

Point 1:

The addition of more recent data improves the manuscript marginally. However the revisions to the article, to my mind, do not adequately address the issues raised initially. "Criticality" is lacking throughout the manuscript. While the discussion section has been revised there is still an underlying weaknesses. In the example below, the first two sentences clearly show a misunderstanding of what constitutes a contribution: devising and exploring a piece of research is not a theoretical contribution.

"5.2 Theoretical implication
The theoretical contributions of this study are as follows. First, the study devised and explored a green economy research framework."

The issues previously raised still remain in the manuscript, and unfortunately there are no easy, quick fixes that can be solved by adding a few sentences or more recent references. A more detailed critical revision of the manuscript is necessary.

Response 1: Thanks for the reviewer’s comment. The author revises the paragraph in the introduction section, and list as follows. Please refer to the revised manuscript in the introduction section. For 2020–2025, the UK Conservative–Liberal Democrat government implemented two key policies. First, the UK government would support innovation that increases the environmental friendliness of products and services. Second, the UK government would encourage resource efficiency and environmental management [1]. The environment–economy nexus is a prominent topic among large companies. The manufacturing industry has placed increasing emphasis on producing environmentally friendly products because of increasing pressure from consumers, environmental activists, and government regulators. The ubiquity of online customer shopping has exerted similar pressure on the e-commerce industry to embrace environmentally conscientious supply chain practices. To meet customer demands for environmentally friendly and sustainable products, firms must establish green-oriented management strategies that emphasize sustainability [2]. With environmental regulations and increasing liability for producing hazardous products, firms are increasingly motivated to incorporate green practices in their operations. Manufacturers are eager to implement strategies such as reverse logistics and recycling to reduce environmental hazards and costs. Firms may be able to effectively combine existing and newly acquired environmental knowledge to gain a competitive advantage over their competitors in e-commerce. Environmental topics such as energy efficiency are also gaining momentum. Numerous firms have complied with environmental regulatory standards by adopting the market-leading Carbon Trust Standard and attaching carbon footprint labels to their products. Environmental and consumer advocates believe in preserving the planet and protecting the environment, and firms must consider these groups when making decisions on green products to improve firm performance.

Thanks for the reviewer’s comment. The author adds the paragraph in the introduction section, and list as follows. Please refer to the revised manuscript in the introduction section. Intrinsic motivation is autonomously activated when people engage in behaviors or activities from which they derive inherent satisfaction, rather than deriving satisfaction by achieving a particular outcome. However, how intrinsic motivation can be applied to eco-friendly living and whether any connection exists between intrinsic motivation and enjoyment in the green context require clarification. If people could satisfy their hedonic needs by demonstrating eco-friendly behaviors, they would be driven by antecedents relating to environmentally friendly behaviors. Marketing managers can thus boost consumers’ desire for and dedication to green living and green consumption by offering environmentally friendly products or services. The positive feedback that people receive from engaging in green consumption enables them to meet their own ethical standards relating to environmental protection.

Thanks for the reviewer’s comment. The author revises the paragraph in the introduction section, and list as follows. Please refer to the revised manuscript in the introduction section. Thus, industries, instead of aiming to enhance consumer enjoyment, must implement environmentally friendly measures to promote “green enjoyment” (GE). As reported by Davis, Bagozzi, and Warshaw [7], enjoyment is a pleasant feeling, and firms must implement strategies to enable consumers to derive enjoyment from their products or services in an environmentally friendly manner.

Thanks for the reviewer’s comment. The author revises the paragraph in the introduction section, and list as follows. Please refer to the revised manuscript in the introduction section. However, questions that remain are whether intrinsic motivation is related to GE and whether the research framework is still reasonable when intrinsic motivation and GE are incorporated. This study proposed that GE is crucial for green consumption and can both increase corporate revenue and support sustainability goals.

Thanks for the reviewer’s comment. The author revises the paragraph in the introduction section, and list as follows. Please refer to the revised manuscript in the introduction section. Green brand love (GBL) is an increasingly critical topic in business. As part of the trend of forming long term relationships with consumers, brand love is prevalent in many different industries. Large companies such as American Eagle Outfitters, Aeropostale, Express, J.Crew, and H&M are contemplating launching brand love initiatives [11]. Moreover, more companies are selling products and services labeled as environmentally friendly, leading to another growing trend. In the 2009 Grocery Manufacturers Association (GMA)/Deloitte Consulting study, GMA noted a major opportunity for companies to meet latent demands for green products. Socially responsible or green goods and services are becoming increasingly important for retailers, and their market presence continues to grow rapidly [12]. Accompanied by appropriate regulation, trade can assist the transition to a green economy by encouraging the exchange of environmentally friendly goods and services. Tully and Winer [12] observed that over half of all participants in their study (60.1%) were willing to pay a premium for such products. Laroche, Michel, Bergeron, Jasmin, and Barbaro-Forleo Guido [13] reported that 80% of consumers who were inclined to pay a premium for green products refused to purchase from companies that do not follow environmental regulations or that misrepresent nongreen products as green. Firms typically announce goals to protect the environment and reduce environmental hazards to both satisfy existing consumers and attract new ones. In terms of personal values, people who actively participate in various environmental protection activities derive satisfaction from living frugally and engaging in environmental protection. Consequently, such consumers are willing to purchase green products or services given their environmental concerns and personal values [14]. Many consumers thus tend to purchase green products in the environmental era [15, 16]. Some companies such as Tesla, Apple, IKEA, and Johnson & Johnson already manufacture products made out of recycled materials to actively promote GBL. Because of its increasingly prominent role, studies have applied GBL as an antecedent construct [17] and outcome construct [17,18], but they have failed to explore the effect of GBL on green purchase intention (GPI).

Thanks for the reviewer’s comment. The author revises the paragraph in the Literature Review and Hypothesis Development section, and list as follows. Please refer to the revised manuscript in the Literature Review and Hypothesis Development section. According to the United Nations’ Intergovernmental Panel on Climate Change, meeting the 1.5°C goal requires reducing carbon emissions to zero by between 2030 and 2050.

Thanks for the reviewer’s comment. The author revises the paragraph in the Literature Review and Hypothesis Development section, and list as follows.

Thanks for the reviewer’s comment. The author adds the paragraph in the Literature Review and Hypothesis Development section, and list as follows. Please refer to the revised manuscript in the Literature Review and Hypothesis Development section. GIM reflects an individual’s tendency to engage in proenvironmental behaviors as well as their interest in, curiosity toward, and self-expression related to green products. For example, a person who purchases more environmentally friendly products because they perceive the products as interesting or because they believe that they can derive satisfaction from performing this proenvironmental behavior rather than perceiving the product as being of value to them is demonstrating the behavior on the basis of intrinsic motivation rather than for extrinsic reasons.

Thanks for the reviewer’s comment. The author revises the paragraph in the Literature Review and Hypothesis Development section, and list as follows. Please refer to the revised manuscript in the Literature Review and Hypothesis Development section. Intrinsic motivation is characterized by personal investment and engagement [33]. Li et al. [26] asserted that with intrinsic motivation

Thanks for the reviewer’s comment. The author revises the paragraph in the Literature Review and Hypothesis Development section, and list as follows. Please refer to the revised manuscript in the Literature Review and Hypothesis Development section. Li et al. [26] noted that GIM reflects an employee’s inherent interest in, love for, passion about, enjoyment of, or satisfaction from proenvironmental behavior. Ali, Ashfaq, Begum, and Ali [34] conducted a study in three cities in China to investigate young Chinese consumers’ behaviors toward purchasing electronic products. They determined that GIM has a positive effect on GPI. However, evidence indicating whether GIM affects GBL is lacking. The present study thus explored the relationship between GIM and GBL to fill the research gap.

Thanks for the reviewer’s comment. The author revises the paragraph in the Literature Review and Hypothesis Development section, and list as follows. Please refer to the revised manuscript in the Literature Review and Hypothesis Development section. Additionally, Park, MacInnis, Priester, Eisingerich, and Iacobucci [41] used the term “brand love emotion” to refer to the specific affective state generated through the consumer–brand relationship.

Thanks for the reviewer’s comment. The author revises the paragraph in the Literature Review and Hypothesis Development section, and list as follows. Please refer to the revised manuscript in the Literature Review and Hypothesis Development section. They used a five-item scale to assess purchase intention.

Thanks for the reviewer’s comment. The author revises the paragraph in the Literature Review and Hypothesis Development section, and list as follows. Please refer to the revised manuscript in the Literature Review and Hypothesis Development section. The present study posited that consumers’ values and interests determine whether consumers are likely to distinguish between environmentally friendly and environmentally unfriendly products and hence whether their GIM engenders an intention to reduce pollution through their purchase behavior. Environmentally unfriendly products include those that do not use recyclable components or appropriate materials for packaging and those manufactured using legal or illegal harmful substances. By contrast, environmentally friendly products include those that use natural substances and sustainable materials and that reduce adverse environmental effects. Lastovica, Bettencourt, Hughner, and Kuntze [50] noted that frugal and eco-centric motivations have positive effects on consumers’ product use behavior. People with high GIM would be expected to purchase eco-friendly products and services. Therefore, this study proposed the following hypothesis:

Thanks for the reviewer’s comment. The author revises the paragraph in the Literature Review and Hypothesis Development section, and list as follows. Please refer to the revised manuscript in the Literature Review and Hypothesis Development section. Waterman [51] stated that feelings of personal expressiveness arise when a person is in the process of self-realization through the fulfillment of their personal potentials, when personal potentials take the form of the “development of one’s skills and talents, the advancement of one’s purpose in living, or both” (p. 679).

Thanks for the reviewer’s comment. The author adds the paragraph in the Literature Review and Hypothesis Development section, and list as follows. Please refer to the revised manuscript in the Literature Review and Hypothesis Development section. For example, Ford offers mass customization options for online consumers, thereby increasing consumers’ enjoyment when they are choosing products. To reduce costs and increase customer responsiveness, firms may develop products that are perceived as environmentally friendly and that can be used to increase GE. Green consumers tend to develop strong relationships with trustworthy brands that provide green products or services. Furthermore, if others perceive green consumers as enjoying green consumption and as gaining satisfaction through interaction with the environment, they may also be attached to an eco-friendly brand. Engaging in an environmentally friendly behavior can also elicit positive emotions through the generation of a positive self-image. Through their relationship with brands, consumers gain opportunities to construct and maintain their identity and to achieve feelings of love and attachment. The more GE they experience, the more positive emotions and enthusiasm they feel. Understanding how to initiate, develop, and maintain high quality consumer–firm relationships is critical to business success [54].

Thanks for the reviewer’s comment. The author revises the paragraph in the Literature Review and Hypothesis Development section, and list as follows. Please refer to the revised manuscript in the Literature Review and Hypothesis Development section. Enjoyment is experienced when needs, including physical, intellectual, and social, are satisfied [51]. GE is a form of happiness that generates hedonic satisfaction through purchase of green products or services. Purchasing is not only about obtaining products or services but is also an enjoyable experience and a form of entertainment [56]. Most studies have reported that enjoyment has a positive effect on purchase intention [55, 57], which increases consumers’ positive attitude toward specific products or services. Consumers especially experience pleasure when they purchase innovative products and support ethical brands, increasing purchase intention. Thus, GE has an effect on both consumers’ attitude and behavioral intention.

Thanks for the reviewer’s comment. The author adds the paragraph in the Literature Review and Hypothesis Development section, and list as follows. Please refer to the revised manuscript in the Literature Review and Hypothesis Development section. If an individual has strong environmental ethics, they may wish to purchase green products to satisfy their hedonic goals. By using green products, consumers perceive that they have contributed to environmental conservation.

Thanks for the reviewer’s comment. The author revises the paragraph in the Literature Review and Hypothesis Development section, and list as follows. Please refer to the revised manuscript in the Literature Review and Hypothesis Development section. Bagozzi, Batra, and Ahuvia [11] maintained that brand love is a crucial topic among brands and marketers. Consumers’ GBL increases their willingness to purchase green products. For example, BMW claims that no two vehicles it manufactures are identical. In this manner, BMW promotes a unique type of brand love of BMW based in its products’ uniqueness, which can strengthen the bond BMW has with its consumers.

Thanks for the reviewer’s comment. The author adds the paragraph in the Literature Review and Hypothesis Development section, and list as follows. Please refer to the revised manuscript in the Literature Review and Hypothesis Development section. GBL increases consumers’ expectancy of product quality and environmental friendliness. Therefore, GBL drives consumers to purchase eco-friendly products with a low environmental impact.

Thanks for the reviewer’s comment. The author revises the paragraph in the methods section, and list as follows. Please refer to the revised manuscript in the methods section. This study used SPSS version 22 (IBM, Armonk, NY, USA) for factor analysis.

Thanks for the reviewer’s comment. The author revises the paragraph in the methods section, and list as follows. Please refer to the revised manuscript in the methods section. Additionally, the questionnaire was distributed over the Internet by using the PTT Bulletin Board System (the largest terminal-based bulletin board system based in Taiwan) to 10 consumers with a minimum of 3 years of green consumption experience.

Thanks for the reviewer’s comment. The author revises the paragraph in the methods section, and list as follows. Please refer to the revised manuscript in the methods section. Tables 1 and 2 present the results of this study and of other related studies. As summarized in the tables, the concepts proposed in this study are innovative and do not overlap with those in related studies. A pretest and factor analysis were conducted on the GIM, GE, GBL, and GPI subscales. According to the exploratory factor analysis results, items 1, 2, and 5 of the GIM subscale had factor loadings of 0.786, 0.860, and 0.691, respectively, and were thus removed. Items 2, 4, 5, and 7 of the GBL subscale had factor loadings of 0.864, 0.861, 0.885, and 0.891, respectively, and were thus removed. Furthermore, item 3 of the GPI subscale had a factor loading of 0.739 and was also removed. The items that had cross-loadings on more than one factor were removed to avoid multicollinearity [65]. This study identified variables relevant to GPI and used them to effectively measure GPI. The definitions and measurements of the constructs in this study are described as follows:

GIM. This study referred to Li et al. [26], who measured intrinsic motivation. To measure GIM, the following six items were used: (1) “I enjoy thinking of new green ideas,” (2) “I enjoy trying to complete environmental tasks in my workplace,” (3) “I enjoy tackling environmental tasks that are completely new,” (4) “I enjoy improving existing green ideas in my workplace,” (5) I become excited when I have new green ideas,” and (6) “I would like to become more engaged in the development of green ideas.”

  1. This study referenced the survey of Tamborini et al. [32] to measure GE by applying the following five items: (1) “The products or services were enjoyable,” (2) “The products or services were entertaining,” (3) “The products or services were appealing,” (4) “The products or services were pleasant,” and (5) “The products or services were fun.”

GBL. This study referred to Salehzadeh et al. [18] for the measurement of GBL, which was performed using the following seven items related to environmental products and services recently used by the respondents: (1) “This is a wonderful green brand,” (2) “This green brand makes me feel good,” (3) “This green brand is amazing,” (4) “This green brand makes me very happy,” (5) “I love this green brand,” (6) “I am passionate about this green brand,” and (7) “I am very attached to this green brand.”

GPI. This study referred to Chen and Deng [49] to measure GPI and applied the following four items: (1) “Purchasing green products is more beneficial than purchasing nongreen products,” (2) “Purchasing green energy-saving products makes me happy,” (3) “When purchasing a product, I consider how it affects the environment,” and (4) “I am willing to spend a little more money to purchase green products.”

Thanks for the reviewer’s comment. The author revises the paragraph in the methods section, and list as follows. Please refer to the revised manuscript in the methods section. E-mail addresses were collected to facilitate contact and were reviewed to ensure that they matched a standard format.

Thanks for the reviewer’s comment. The author revises the paragraph in the methods section, and list as follows. Please refer to the revised manuscript in the methods section. A total of 302 completed questionnaires were retrieved from 386 distributed questionnaires; the response rate was 78.24%. This study adopted convenience sampling to collect the data samples. Convenience sampling enables the researcher to freely choose sample group members. Nonprobability sampling lacks the advantage of every data sample of a particular size having an equal chance of being selected, but this is often unavoidable. Because online data collection is quick and cost-effective, many researchers have embraced this nonrepresentative method. Harvard’s Project Implicit, which offers implicit-association tests, is one example. The nonprobability sampling procedure relies on the person conducting the sampling, which can elicit other complicated concerns. This person conducting the sampling must be knowledgeable of the population and phenomena being studied [67]. However, simple random sampling is a favored method for achieving sufficient external validity. Moreover, a larger sample size is required to eliminate the margin of error.

Thanks for the reviewer’s comment. The author revises the paragraph in the results section, and list as follows. Please refer to the revised manuscript in the results section. This study used descriptive statistics to describe the basic features of the study data. Table 3 presents the correlations between the constructs and indicates that the means and standard deviations followed normal distributions.

Thanks for the reviewer’s comment. The author revises the paragraph in the results section, and list as follows. Please refer to the revised manuscript in the results section. The variance inflation factor values of the exogenous constructs were all less than 5 and are listed in Table 4. This study had no multicollinearity problem [68], and the results exhibited acceptable reliability and validity.

Thanks for the reviewer’s comment. The author revises the paragraph in the results section, and list as follows. Please refer to the revised manuscript in the results section. Table 5 presents the factor loading, AVE, and construct reliability results. The AVE values for GIM, GE, GBL, and GPI were 0.58, 0.67, 0.62, and 0.54, respectively, which all exceed 0.5, thus indicating acceptable convergent validity.

Thanks for the reviewer’s comment. The author revises the paragraph in the results section, and list as follows. Please refer to the revised manuscript in the results section. This study used the PROCESS macro version 2.15 to test the mediating effect of GE on the relationship between GIM and GBL; the effect sizes were 0.059 and 0.250, respectively. Harman’s single-factor test was used to avoid common method bias. The variance value in this test was 40.94%, which was less than the threshold of 50%. Therefore, this study had no common method bias problem.

Thanks for the reviewer’s comment. The author revises the paragraph in the results section, and list as follows. Please refer to the revised manuscript in the results section. In the explicit statistical test of the measurement model and structural model, the chi-square difference test revealed that χ2 (11.9, 3) > 11.34 at the 0.01% significance level. Thus, this study conducted an assessment of the hypothesized paths in the structural model. As presented in Table 6, he path analysis revealed that GIM influenced GE (t = 9.15), thus supporting H1. GIM had a positive direct effect on GBL (t = 7.81) and GPI (t = 3.73); thus H2 and H3 are fully supported. This finding is consistent with that of Ali et al. [34]. GIM had a positive effect on GPI. Moreover, GIM had a significant total effect on GPI when influenced by GE and GBL (t = 5.35). GE positively influenced GBL (t = 5.39) and GPI (t = 2.60), thus supporting H4 and H5. H6, which referred to the relationship between GBL and GPI, was also supported. Therefore, GIM, GE, GBL, and GPI were positively correlated and GBL positively affected GPI.

Thanks for the reviewer’s comment. The author revises the paragraph in the results section, and list as follows. Please refer to the revised manuscript in the results section. The original figure was exported from the statistical software and is presented in Appendix A.

Thanks for the reviewer’s comment. The author adds the paragraph in the conclusion section, and list as follows. The practical contributions of this study are as follows. Please refer to the revised manuscript in the conclusion section. Although enjoyment is theoretically rooted in pleasure and pain, feelings of personal expressiveness are conceptually linked with feelings associated with intrinsic motivation [77]. GE is generated alongside satisfaction with and interest in green products or services.

Thanks for the reviewer’s comment. The author adds the paragraph in the conclusion section, and list as follows. The practical contributions of this study are as follows. Please refer to the revised manuscript in the conclusion section. GBL are rarely analyzed with respect to its antecedent and outcome variables. Understanding GBL in purchase behaviors could be helpful for the brand managers in creating stronger brands. This study depicts GIM and GE as drivers of GBL which is relatively new. The proposed GPI model also proves that GBL reinforces the effect of brand engagement on purchase attitude.

Thanks for the reviewer’s comment. The author adds the paragraph in the conclusion section, and list as follows. The theoretical contributions of this study are as follows. Please refer to the revised manuscript in the conclusion section. The theoretical contributions of this study are as follows. First, the idea of increasing GIM, GE, GBL, and GPI represents a novel and innovative strategy. Second, although the concepts of GBL and GPI have been examined in other studies, the present study proposes the new constructs GIM and GE and analyzed the relationships among GIM, GE, GBL, and GPI to fill the literature gap. Third, this study integrated the concepts of GIM, GE, GBL, and GPI into a research framework of GPI to extend green economy research. Fourth, Salehzadeh et al. [18] determined that green brand image affects green brand attitude, GBL, and green trust but failed to explore the relations between GBL and other variables. This study incorporated GIM, GE, GBL, and GPI into a new model to expand the relevant literature.

Thanks for the reviewer’s comment. The author adds the paragraph in the conclusion section, and list as follows. The theoretical contributions of this study are as follows. Please refer to the revised manuscript in the conclusion section. Therefore, the antecedent of GBL is GIM, and the consequence of GBL is GPI; the partial mediators are GE and GBL. GIM can directly affect GPI, and the influences of GE and GBL are also crucial. Therefore, firms can enhance their GBL through the promotion of GIM and GE. This study also identified the mediating role of GE and GBL in the relationship between GIM and GPI, such that the strong effects of GIM on GPI occur only when GE and GBL are emphasized but are weaker when GIM is emphasized alone. Most research has focused on GIM, without attempting to explore other factors. By expressly focusing on three dimensions, the present study identified the factors that drive GPI. Integrating the new concept of GE is a key step in creating, marketing, and presenting a green brand. The introduction of GE in this study, specifically in the field of green business, has changed the interpretation of the relationship between GE and GBL. The study employed the definition of enjoyment provided by Tamborini et al. [32] together with GIM and GBL to propose the research model in which GIM and GE influence GBL and GPI, which shapes GBL and GPI as consumer responses. Therefore, the mediating role of GE in the relationship between GIM and GBL was revealed. Similar studies [17, 18] have failed to explore the mediating role of GBL. Thus, this study has advanced the GBL literature by exploring its antecedent and outcome variables. The results can serve as a reference for proposing theories related to GBL and GE. This study also contributed to the literature by identifying the relationships among GIM, GE, and GBL, thereby advancing green business research.

Thanks for the reviewer’s comment. The author adds the paragraph in the conclusion section, and list as follows. The limitation of this study are as follows. Please refer to the revised manuscript in the conclusion section. The present study includes some limitations. First, the cross-sectional approach limits the ability to make causal inferences about the findings. Second, online survey may lead to biased data which limits the statistical power.
